# Combining genome-wide association studies highlight novel loci involved in human facial variation

Ziyi Xiong [1,2,9], Xingjian Gao[3,4,9], Yan Chen [1,3,9], Zhanying Feng [5], Siyu Pan[3], Haojie Lu[2,6], Andre G. Uitterlinden [2,6], Tamar Nijsten[7], Arfan Ikram [2], Fernando Rivadeneira [2,6,8], Mohsen Ghanbari [2], Yong Wang [5], Manfred Kayser [1,10] ✉ & Fan Liu [1,3,10] ✉

Standard genome-wide association studies (GWASs) rely on analyzing a single trait at a time. However, many human phenotypes are complex and composed by multiple correlated traits. Here we introduce C-GWAS, a method for combining GWAS summary statistics of multiple potentially correlated traits. Extensive computer simulations demonstrated increased statistical power of C-GWAS compared to the minimal $p$-values of multiple single-trait GWASs (MinGWAS) and the current state-of-the-art method for combining single-trait GWASs (MTAG). Applying C-GWAS to a meta-analysis dataset of 78 single trait facial GWASs from 10,115 Europeans identified 56 study-wide suggestively significant loci with multi-trait effects on facial morphology of which 17 are novel loci. Using data from additional 13,622 European and Asian samples, 46 (82%) loci, including 9 (53%) novel loci, were replicated at nominal significance with consistent allele effects. Functional analyses further strengthen the reliability of our C-GWAS findings. Our study introduces the C-GWAS method and makes it available as computationally efficient open-source R package for widespread future use. Our work also provides insights into the genetic architecture of human facial appearance.

Standard genome-wide association studies (GWASs) rely on analyzing a single trait at a time. However, many human phenotypes are complex and composed of multiple correlated traits. A common approach in dealing with correlated traits in GWAS is to separately conduct GWASs for the different traits and use the minimum $p$-values of multiple GWASs (MinGWAS) with a more stringent significance threshold.

However, because MinGWAS cannot exploit genetic information underlying correlated traits, it is of limited power in detecting genetic variants with multi-trait effects.

Several methods were previously developed to enable integrated analysis of multiple traits based on single-trait GWAS summary statistics[1–3], which are more flexible than dimension reduction of

[1]Department of Genetic Identification, Erasmus MC, University Medical Center Rotterdam, Rotterdam, the Netherlands. [2]Department of Epidemiology, Erasmus MC, University Medical Center Rotterdam, Rotterdam, the Netherlands. [3]CAS Key Laboratory of Genomic and Precision Medicine, Beijing Institute of Genomics, Chinese Academy of Sciences, Beijing, China. [4]National Clinical Research Center of Kidney Diseases, Jinling Hospital, Nanjing, Jiangsu, China. [5]CEMS, NCMIS, HCMS, MDIS, Academy of Mathematics and Systems Science, Chinese Academy of Sciences, Beijing, China. [6]Department of Internal Medicine, Erasmus MC, University Medical Center Rotterdam, Rotterdam, the Netherlands. [7]Department of Dermatology, Erasmus MC, University Medical Center Rotterdam, Rotterdam, the Netherlands. [8]Department of Oral and Maxillofacial Surgery, Erasmus MC, University Medical Center Rotterdam, Rotterdam, the Netherlands. [9]These authors contributed equally: Ziyi Xiong, Xingjian Gao, Yan Chen. [10]These authors jointly supervised this work: Manfred Kayser, Fan Liu. ✉e-mail: m.kayser@erasmusmc.nl; f.liu@erasmusmc.nl

individual-level data. Such methods have shown increased power than MinGWAS in detecting genetic associations, but have limitations. The most prominent example is the current state-of-the-art method for integrated GWAS named multi-trait analysis of GWAS (MTAG)[3]. MTAG distinguishes true effect and estimate error and obtains higher power than other competing methods. However, MTAG relies on the key assumption that all SNPs share the same variance–covariance matrix of effect sizes across all of the traits tested. Both, the statistical power and the type-I error rate ($\alpha$) of MTAG are sensitive to the parameters related to this assumption. Moreover, accurate estimation of these parameters requires high-powered input GWASs. When the assumption is satisfied but the input GWASs do not have sufficiently high statistical power, which often is barely achievable with tens or even hundreds of thousands of samples, MTAG is expected to have limited power and unstable $\alpha$. Furthermore, the key assumption of MTAG is violated as soon as different SNPs are associated with different traits of different effect sizes, which is likely the case for many if not all complex traits. When this assumption is violated, MTAG also has limited power and unstable $\alpha$.

Aiming to overcome the limitations of previous methods, we developed C-GWAS, a method to combine GWAS summary statistics of multiple potentially correlated traits. We implemented C-GWAS as an R package that can effectively handle a large number of traits in a parallel manner by optimizing load splitting and memory management. We performed extensive computer simulations under a variety of scenarios to assess the performance of C-GWAS in comparison with MinGWAS and MTAG. To empirically exemplify its practical suitability, we applied C-GWAS to a previously published summary statistic dataset of 78 GWASs of correlated facial traits with mediocre statistical power ($N$ = 10,115 Europeans)[4] and compared it with MTAG and other competing methods. We used new and previously published data ($N$ = 13,622 Europeans and Asians)[5,6] for replicating the C-GWAS findings and accumulated functional evidence to further strengthen the reliability of the novel genetic face loci we discovered with C-GWAS.

## Results

### Overview of C-GWAS

C-GWAS represents a complete solution for investigating potential multi-trait effects of SNPs based only on the summary statistics of $K$ GWASs (Supplementary Fig. 1). For optimizing statistical power, it is necessary to partition the correlation matrix of the $T$ statistics from $K$ GWASs into two matrices that differentiate the 'effect correlation' $\mathbf{\Pi}$ caused purely by true allelic effects from the 'background correlation' $\mathbf{\Psi}$ under the null. Both $\mathbf{\Psi}$ and $\mathbf{\Pi}$ are $K$ by $K$ symmetric matrices, where $\Psi_{ij}$, $i \in \{1, 2,..., K\}$ and $j \in \{1, 2,..., K\}$ (in brief $\psi$), indicating the correlation between the $T$ statistics of $i$th GWAS and $j$th GWAS caused only by non-genetic effects and similarly $\Pi_{ij}$ (or $\pi$) indicating the correlation caused only by true allelic effects. In standard meta-analysis, the off-diagonal elements of $\mathbf{\Psi}$ would be zero. In C-GWAS, however, the off-diagonal of $\mathbf{\Psi}$ may deviate from zero in presence of sample overlap and/or non-genetic phenotype correlations caused by shared environmental or unknown factors. C-GWAS and MTAG have an important similarity in distinguishing $\mathbf{\Pi}$ from $\mathbf{\Psi}$ as key parameters in combining GWASs. Note that the true effect $\mathbf{\Omega}$ and the estimate error $\mathbf{\Sigma}$ in MTAG are also $K$ by $K$ symmetric matrices and ideally, $\Pi_{ij} = \Omega_{ij}/\sqrt{\Omega_{ii}\Omega_{jj}}$ and $\Psi_{ij} = \Sigma_{ij}/\sqrt{\Sigma_{ii}\Sigma_{jj}}$. Different from MTAG that generates genome-wide $p$-value vectors with the same number of input GWASs, C-GWAS generates a single vector of genome-wide $p$-values testing for each SNP if the null is deviated (H0: absence of any effect on all traits, H1: deviation from zero for at least one trait). In addition, different from MTAG that performs joint analysis of all GWASs at once, C-GWAS use two-step design where separately employs an iterative decision-making algorithm and truncated test strategy to take into account that SNPs may have different effects in different subsets of GWASs.

**Two-step design.** Full details regarding formulation, derivation, and simulation can be found in the "Methods" section. In brief, C-GWAS uses a two-step design to achieve high power in detecting multi-trait effects under various complex scenarios with the emphasis that SNPs may have different effects in different subsets of potentially correlated GWASs. The first step is to use iterative Effect-based Inversed Covariance Weighting (i-EbICoW) to combine subsets of $K$ GWASs in which $\mathbf{\Pi}$ can be distinguished from $\mathbf{\Psi}$. EbICoW is originated from traditional Inversed Variance Weighting (IVW) based meta-analysis with three extensions. For each SNP, let $\mathbf{\sigma} = (\sigma_1, \sigma_2, \ldots, \sigma_K)^{\mathrm{T}}$, $\mathbf{n} = (n_1, n_2, \ldots, n_K)^{\mathrm{T}}$ and $\mathbf{t} = (t_1, t_2, \ldots, t_K)^{\mathrm{T}}$ be the vectors of standard errors, sample sizes, and $T$ statistics from $K$ GWASs for one SNP, the combined $T$ statistic of IVW is expressed as $t_{meta} = \mathbf{w}^{\mathrm{T}}\mathbf{t}/\sqrt{\mathbf{w}^{\mathrm{T}}\mathbf{w}}$, where $\mathbf{w}$ is a vector of weights, which are usually chosen from $\mathbf{w} = (\sqrt{n_1}, \sqrt{n_2}, \ldots, \sqrt{n_K})^{\mathrm{T}}$ or $\mathbf{w} = (1/\sigma_1, 1/\sigma_2, \ldots, 1/\sigma_K)^{\mathrm{T}}$. The first extension of EbICoW guarantees the standard normal distribution under the null for combining correlated GWASs, i.e., the presence of non-zero off-diagonal elements of $\mathbf{\Psi}$. The revised combined $T$ statistic can be derived as a function of $\mathbf{\Psi}$ and $\mathbf{w}$ and $\mathbf{t}$ from the original IVW.

$$t_1 = \frac{\mathbf{w}^{\mathrm{T}}\mathbf{\Psi}^{-1}}{\sqrt{\mathbf{w}^{\mathrm{T}}\mathbf{\Psi}^{-1}\mathbf{w}}}\mathbf{t} \tag{1}$$

The second extension optimizes the statistical power when elements in the vector of expected true effects $\mathbf{\delta} = (\delta_1, \delta_2, \ldots, \delta_K)^{\mathrm{T}}$ are different. Let $\mathbf{b}$ be a vector of 1 with length $K$, and $\mathbf{H}$ be a $K$ by $K$ covariance matrix of $\mathbf{\delta}$ among the genome-wide SNPs, where $H_{ij} = \mathrm{E}(\delta_i \delta_j)$.

$$t_2 = \frac{(\mathbf{b}^{\mathrm{T}}\mathbf{H} \circ \mathbf{w}^{\mathrm{T}})\mathbf{\Psi}^{-1}}{\sqrt{(\mathbf{b}^{\mathrm{T}}\mathbf{H} \circ \mathbf{w}^{\mathrm{T}})\mathbf{\Psi}^{-1}(\mathbf{b}^{\mathrm{T}}\mathbf{H} \circ \mathbf{w}^{\mathrm{T}})^{\mathrm{T}}}}\mathbf{t} \tag{2}$$

The third extension accounts for different effect directions of the expected $\mathbf{\delta}$. For $k \in \{1, 2,..., K\}$,

$$t_3 = \frac{(\mathrm{sign}(\mathbf{\Pi} - \mathbf{\Psi})_k \mathbf{H} \circ \mathbf{w}^{\mathrm{T}})\mathbf{\Psi}^{-1}}{\sqrt{(\mathrm{sign}(\mathbf{\Pi} - \mathbf{\Psi})_k \mathbf{H} \circ \mathbf{w}^{\mathrm{T}})\mathbf{\Psi}^{-1}(\mathrm{sign}(\mathbf{\Pi} - \mathbf{\Psi})_k \mathbf{H} \circ \mathbf{w}^{\mathrm{T}})^{\mathrm{T}}}}\mathbf{t} \tag{3}$$

here the function *sign* for a variable $x$ is defined as $\mathrm{sign}(x) = 1$ when $x \geq 0$ and $\mathrm{sign}(x) = -1$ when $x < 0$, so that for a matrix $\mathbf{X}$, $[\mathrm{sign}(\mathbf{X})]_{ij} = \mathrm{sign}(X_{ij})$. Note that Eqs. (1), (2), and (3) are all equivalent to the original form of IVW when all GWASs are uncorrelated and all expected $\mathbf{\delta}$ are the same (see "Methods").

C-GWAS then iteratively applies a decision-making algorithm to find and combine subsets of $K$ GWASs that are suitable for EbICoW, i.e., when off-diagonal elements of $\mathbf{\Pi}$ are difficult to be distinguished those from $\mathbf{\Psi}$. It iteratively combines a pair of GWASs with the max $|\pi - \psi|$ until none of the GWAS pairs is suitable for EbICoW.

The second step is to use Truncated Wald Test (TWT) to combine the revised GWASs in a SNP-specific manner. For each SNP, TWT identifies its most significant subset and produces a single combined $p$-value by applying the Wald test[1] (Wald) to all subsets satisfying a series of preset thresholds. Different from i-EbICoW, in which the $T$ statistics asymptotically follow the standard normal distribution, TWT results obtained without calibration do not follow any known distribution due to procedures for selecting SNP-specific subset.

**Parameter estimation and distribution calibration.** Coming with C-GWAS, three methods were additionally developed as functions for estimating inflation (*getI*), $\psi$ (*getPsi*), and for calibration of unknown distributions (*getCoef*). The *getI* removes potential inflations of input GWASs utilizing the relationship between the distribution of the true effect-excluded GWAS and the standard normal distribution. The *getPsi* partitions the correlation matrix of GWASs into $\mathbf{\Psi}$ and $\mathbf{\Pi}$ in such

a way that $\psi$ is estimated as the correlation of two GWASS by only focusing on non-significant SNPs in the joint distribution of the two GWASS. The *getCoef* function calibrates TWT *p*-values according to quantile specific coefficients between an empirically simulated null and the uniform distribution. This guarantees that final *p*-values from C-GWAS are directly comparable with those from any standard GWAS.

## C-GWAS performance via computer simulations

Extensive computer simulations were carried out to assess the performance of C-GWAS in terms of statistical power (scenarios 1–5), parameter estimation (scenarios 6-7), and type-I error (scenarios 8–10), in comparison with MTAG and MinGWAS whenever appropriate.

Scenario 1 is designed to compare the power between two intermediate forms of EbICoW (Eqs. 1 and 2) for combining summary statistics of two GWASS (Supplementary Fig. 2). The simulations confirm that (1) is a special case of (2) when the true effects and the sample sizes are the same between the two GWASS. When they are different, the power of (2) is almost always higher than that of (1). The larger the |

$\pi-\psi|$ is, and the larger the difference of true effects, the larger the power of (2), particularly when the sample sizes are the same. These results demonstrate that (2) has a higher power than (1) in most circumstances.

Scenario 2 compares the power between an intermediate (2) and the final form of EbICoW (3) for combining summary statistics of two GWASS (Supplementary Fig. 3). The simulations showed that when $\pi>\psi$, the power of (3) is the same as (2), and when the $\pi<\psi$, the power of (3) is the same as negative signed (2), which has a higher power than (2). The power gain is increasingly considerable when the true effects become large. These results demonstrate that (3) has the same or higher power than (2).

Scenario 3 explores the power relationship between EbICoW and Wald under different $\pi$ and $\psi$ when combining summary statistics of two GWASS (the central panel of Fig. 1 and Supplementary Fig. 4). Simulations showed that the power of both EbICoW and Wald increase with $|\pi-\psi|$, but the power of EbICoW increases in a faster rate than that of Wald. Hence, the power of EbICoW is larger than that of Wald when $|\pi-\psi|$ is large. In contrast, the power of Wald is higher than that of

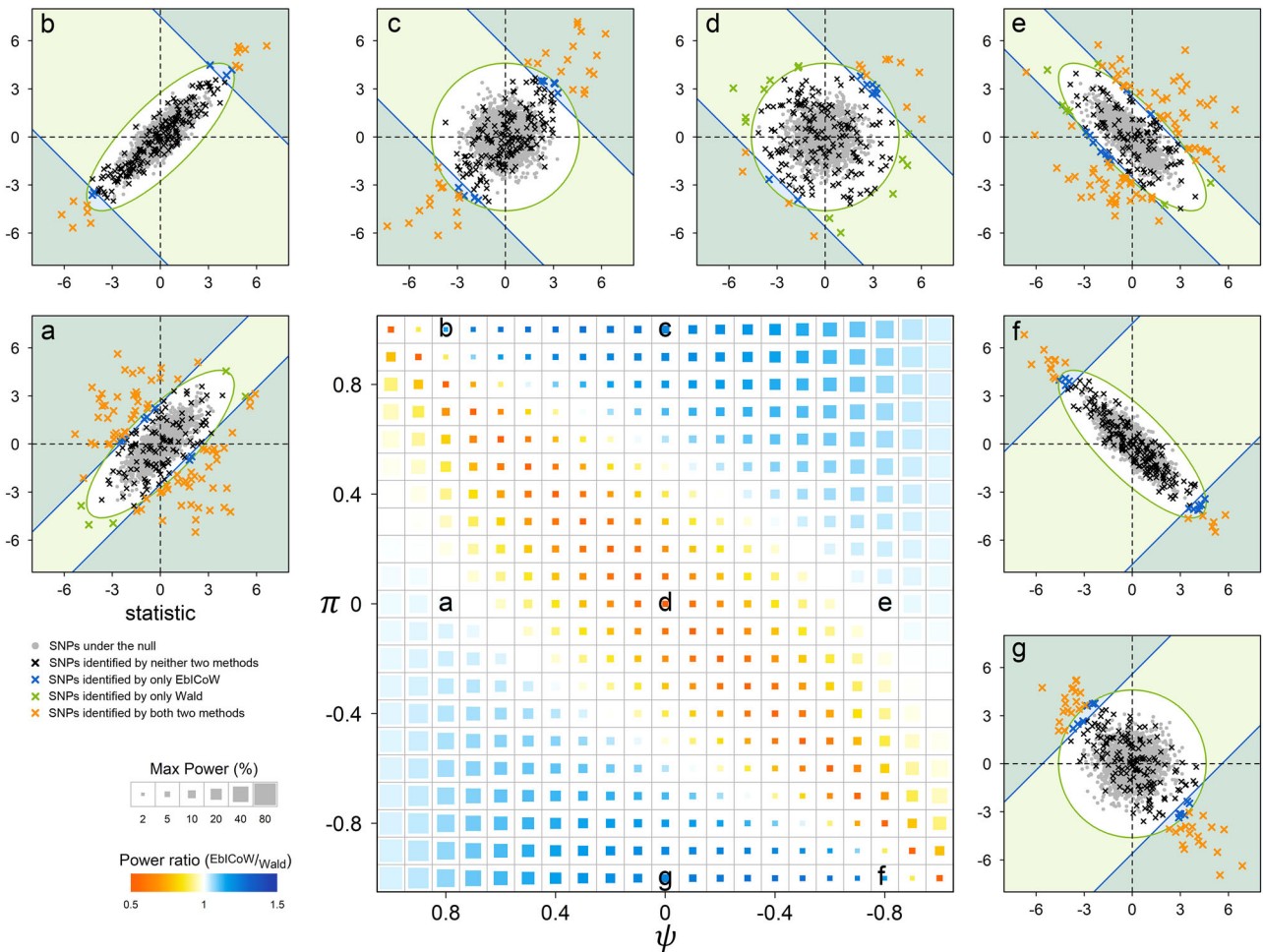

**Fig. 1 | Power comparison between EbICoW and Wald test for combining two GWASS under the same true effect via simulations.** The summary statistics of two GWASS with the same true effect configuration, i.e., E($\chi^2$) = 1.3, were simulated, each consisting of 50,000 SNPs with 10% having true effect, according to the following parameter grid: $\psi$ and $\pi\in\{-0.98, -0.9, -0.8,...,-0.1, 0, 0.1,..., 0.8, 0.9, 0.98\}$. For each of the 441 (21×21) different combinations of $\pi$ and $\psi$, 1000 replicates were carried out. The power relationship between EbICoW and Wald is illustrated using a heat map in the center panel. The maximum power of the two methods is proportional to the block size in the heat map. The power ratio of EbICoW and Wald is expressed by color from red to blue. Taking seven specific combinations of $\pi$ and $\psi$ as examples, we show in detail the performance of EbICoW and Wald (panels (**a**–**g**) surrounding the central panel). For illustration purposes, each surrounding panel is a scatter plot of the test statistics from the two GWASS consisting of 2000 SNPs with 50% increased true effect compared with the corresponding cell in the heat map, in which the null SNPs are indicated in gray dots and the SNPs with true effects are indicated in colored crosses. Different colors are used to distinguish the significance of EbICoW and Wald, i.e., orange for both significant, blue for EbICoW only significant, green for Wald only significant, and black for non-significant results. The areas outside of the blue lines are the detection range of EbICoW and the area outside of the green ellipse is the detection range of Wald.

EbICoW when $|\pi-\psi|$ is small, especially when two GWASS have the same true effect. The statistical nature of EbICoW and Wald when combining two GWASS is illustrated in panels (a–g) of Fig. 1 and Supplementary Fig. 4. Notably, Wald only uses the joint distribution of null to determine its detection range, where the null is determined only by $\psi$. In contrast, EbICoW uses the joint distributions of both null and true effects to determine its detection range, which are determined by both $\pi$ and $\psi$. This explains the power gain of EbICoW relative to Wald when $|\pi-\psi|$ increases.

Scenario 4 compares the power between TWT, Wald, and Min-GWAS for combining multiple GWASS (here we simulated ten GWASS) under the same $\alpha$ (Supplementary Fig. 5). Simulations showed that TWT has a similar or higher power than Wald and MinGWAS in all scenarios investigated. One extreme scenario is the presence of only one GWAS having a true effect and the absence of background correlations. Under this scenario, the power of TWT is almost the same as MinGWAS. Here the loss of power due to the extra burden of distribution calibration for TWT is negligible. Another extreme scenario is

the presence of equally distributed true effect for all GWASS and the absence of background correlations. Under this scenario, the power of TWT is almost the same as Wald. For all other simulated scenarios, TWT has the optimal power. A general pattern is that compared with Wald, the gain of power in TWT decreases with the increase of $\pi$ and increases with the increase of $\psi$.

Scenario 5 compares the power between EbICoW, TWT, MTAG and MinGWAS for combining two GWASS (Supplementary Fig. 6) as well as for combining multiple GWAS (here we simulated ten GWASS, Fig. 2) under the same $\alpha$. For combining two GWASS, a general pattern is that when $|\pi-\psi|=0$, the power of TWT > MTAG = MinGWAS > EbICoW, and when $|\pi-\psi|$ becomes large, the power of EbICoW > MTAG > TWT > MinGWAS. The exception is that when the difference of true effect between two GWASS is large ($E(\chi^2_{1|2})=1.1|1.6$), the power of EbICoW is always the highest, regardless of $|\pi-\psi|$. For combining multiple GWASS, we observed the following patterns. First, MinGWAS almost always performs the worst. Second, the power of MTAG is not the highest at any time, either lower than EbICoW or lower than TWT.

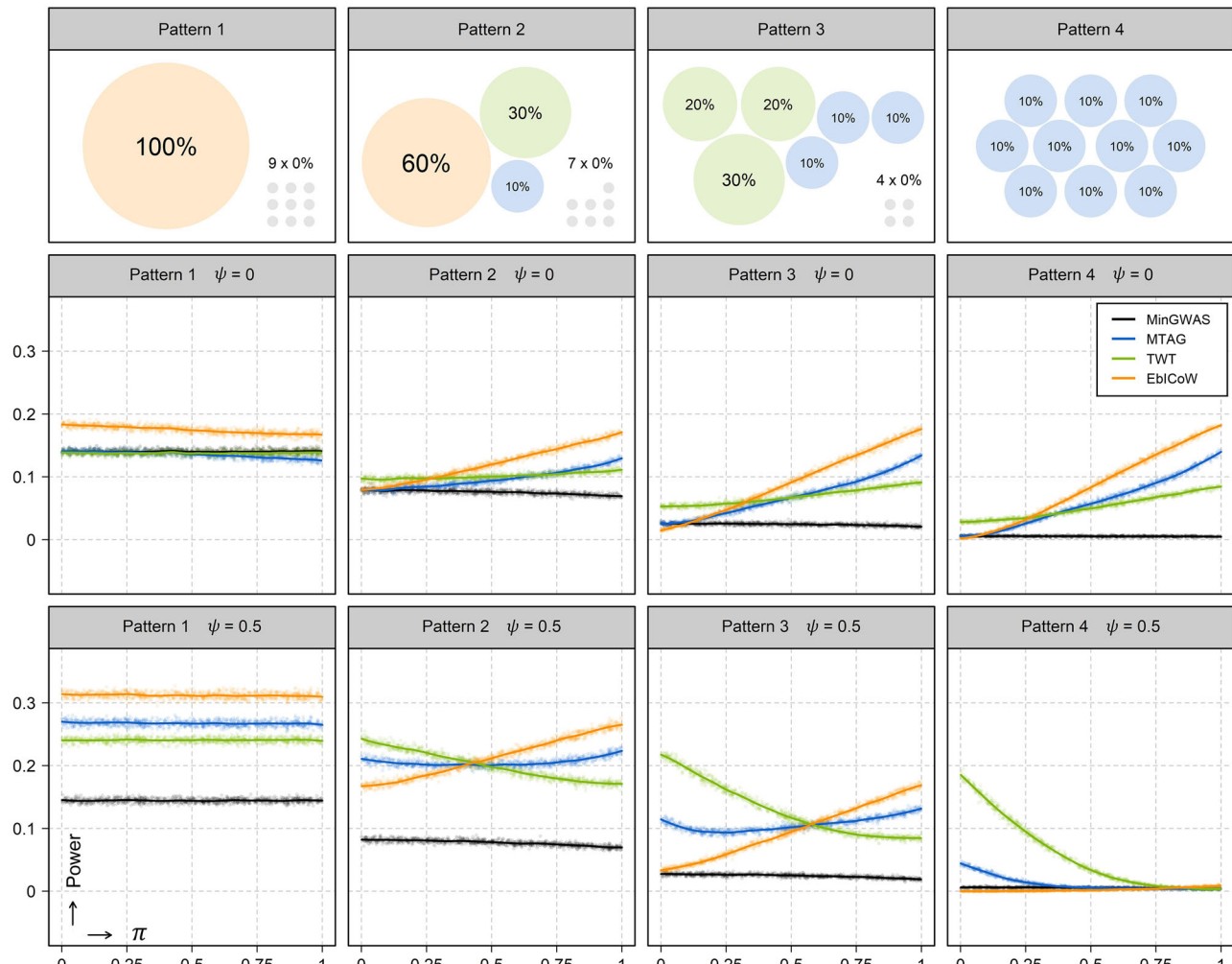

**Fig. 2 | Power comparison between EbICoW, MTAG and TWT via simulations.** The summary statistics of ten GWASS were simulated, each consisting of 50,000 SNPs with 10% having true effect. We evaluated the power of EbICoW, MTAG, TWT and MinGWAS at $\alpha=0.05/50,000$ under two different $\psi \in \{0, 0.5\}$ and four different patterns of true effect. For pattern 1, one GWAS with true effect $E(\chi^2)=2$ and nine GWASS under the null $E(\chi^2)=1$ were simulated; for pattern 2, three GWASS with true effect $E(\chi^2)\in\{1.6, 1.3, 1.1\}$ and seven GWASS under the null were simulated; for pattern 3, six GWASS with true effect $E(\chi^2)\in\{1.3, 1.2, 1.2, 1.1, 1.1, 1.1\}$ and four GWASS under the null were simulated; for pattern 4, all ten GWASS were simulated with true effect $E(\chi^2)=1.1$. For each of the 8 different combinations of $\psi$ and pattern of true

effect, 1,000 replicates were carried out. The x-axis is $\pi$ and the y-axis is the power. Each column has the same configuration of the true effect and each row has the same $\psi$. The configuration of the patterns of true effect is displayed in the top row. Note that the $E(\chi^2)$ is the sum of mean $\chi^2$ of true effect and the mean $\chi^2$ under the null, the latter is always 1.0. The power estimates of EbICoW, TWT, MTAG, and MinGWAS of ten GWASS are indicated in orange, green, blue, and black dots, respectively. Dots were fitted using local polynomial regression curves. Note that the p-values from MTAG, TWT, and the MinGWAS of ten GWASS were additionally adjusted to have the same $\alpha$ with EbICoW.

Third, when $\pi$ is large, EbICoW performs the best and when $\pi$ is small, TWT performs the best. Fourth, when $\psi \neq 0$, the relative power of TWT increases with the proportion of GWASs with true effects. The exception is the pattern 1, i.e., when only one GWAS has a true effect, in which case EbICoW always performs the best regardless of other parameters. These results convincingly demonstrate that a combined use of EbICoW and TWT as done in C-GWAS outperforms MTAG and MinGWAS, which also likely holds true when comparing C-GWAS with any statistical approach that is originated purely from the IVW or the Wald.

Scenario 6 assesses the performance of our function for estimating inflation (*getI*, Supplementary Fig. 7). Simulations showed that *getI* always produces unbiased estimation, although the estimation error negatively correlates with the size of true effect and positively correlates with the proportion of SNPs having true effects. On the other hand, it is obvious that the genomic control method produces an increasing bias with the increase of the true effect and with the increase of the proportion of SNPs having true effects.

Scenario 7 assesses the performance of the function for estimating $\psi$ (*getPsi*, Supplementary Fig. 8). Simulations showed that *getPsi* always produces unbiased estimation of $\psi$ with very high accuracies. Simply computing the correlation between two GWASs using all test statistics[1] is not only biased but also has large estimation errors, especially when true effects are large. Computing the correlation between two GWASs by only focusing on nominally non-significant SNPs in their marginal distribution[7] also produces biased results, i.e., it tends to over-estimate $\psi$ when $\pi < 0$ and under-estimate $\psi$ when $\pi > 0$.

Scenario 8 explores the $\alpha$ of all intermediate results from C-GWAS, i.e., results from EbICoW and TWT, under three different levels of estimation errors of *getI* and *getPsi* (Supplementary Fig. 9). These levels were ascertained according to our simulation results from Scenario 6 and Scenario 7 to cover the whole range of estimation errors. Simulations showed that the study-wide $\alpha$ of both EbICoW and TWT are stable at 5% or slightly higher than 5%. Larger estimation errors of inflation and $\psi$ indeed led to an increased $\alpha$, with a slightly pronounced effect for EbICoW. In the absence of estimation errors, $\alpha$ is always stable at 5%, while in the presence of estimation errors, the increase of the number of phenotypes slightly increased the study-wide $\alpha$. These results demonstrate that the intermediate results of C-GWAS (EbICoW and TWT) have highly stable study-wide $\alpha$.

Scenario 9 evaluates the performance of *getCoef* in adjusting for the final *p*-values from C-GWAS (Fig. 3). Simulations confirmed that the final *p*-values produced by *getCoef* always follow the uniform distribution under the null, regardless to the $\Psi$ and regardless to the shape of the initial distribution. In multiple-trait studies, a commonly used approach for multiple testing correction is to adjust for the number of independent tests, which can be derived either analytically using the matrix spectral decomposition analysis[8] or empirically using simulations[5]. Our simulations showed that using the number of independent tests guarantees a uniform distribution under the null only for correcting MinGWAS without background correlations, but is slightly biased at least in the presence of background correlations leaving alone other types of unknown initial distributions.

Scenario 10 assesses the performance of C-GWAS and MTAG in combining GWASs of overlapping samples under extreme settings, i.e., we attempt to combine three GWASs of the same phenotype, which are conducted in fully overlapping, partially overlapping, or non-overlapping sub-samples. Our simulations showed that under all investigated settings, the test statistics from C-GWAS for combining the three sub-sample GWASs tightly followed those from the GWAS conducted in the union of three sub-samples (Supplementary Fig. 10). Instead of providing a single vector of combined test statistics as done in C-GWAS, MTAG provides three sets of combined test statistics for combining the three GWASs. In fully overlapping samples, the results of MTAG also tightly followed those from the GWAS. When the sample overlapping substantially reduces, MTAG results slightly deviated

from GWAS results, as explained by increased errors in estimating $\psi$ and $\pi$ (Supplementary Fig. 10).

## C-GWAS R package and computational efficiency

C-GWAS was implemented as a user-friendly, publicly accessible, OS-independent, and parallel R package called "C-GWAS", https://github.com/Fun-Gene/CGWAS. We assessed the performance of C-GWAS and compared it with MTAG using simulated data of 6 million SNPs; 5, 10, 20, 30, 40, and 80 GWASs; and 1, 2, 4, 8, and 16 paralleled threads. C-GWAS appeared more computationally effective than MTAG as indicated by several patterns (Supplementary Fig. 11). First, C-GWAS utilizes multi-threads in parallel to reduce computational time, while MTAG only runs in serial. Second, the peak memory usage of C-GWAS becomes much smaller than that of MTAG when the number of GWASs increases e.g., exceeding 20. Third, the computational time of C-GWAS become much less than MTAG when the number of GWASs exceeds 20, already under the single-thread setting. Fourth, MTAG becomes impractical when the number of GWASs exceeds 30 due to out of memory (>256 GB), while C-GWAS completed the analysis of 80 GWASs in 1.3 hours with 16 threads in parallel and peak memory usage of 32 GB.

## C-GWAS outperforms MinGWAS in real data of the human face

To empirically demonstrate the power gain of C-GWAS indicated in the simulations, we applied C-GWAS to real data for which MinGWAS have been conducted previously[4], thereby allowing direct method comparison in the very same dataset. Human facial variation represents a multidimensional set of correlated and mostly symmetric traits with high heritability. The previous MinGWAS was conducted in 10,115 samples from four European cohorts[4], which involved 78 GWASs of different facial traits representing pair-wise distances between 13 anatomically meaningful facial landmarks (Fig. 4a). The previous MinGWAS had identified 6 study-wide significant loci and 24 study-wide suggestively significant loci[4].

For a fair comparison, we adjusted the previous MinGWAS, which is by definition inflated under the null, using our *getCoef* function. A simulation analysis imbedded in C-GWAS confirmed that under the null, both C-GWAS and the adjusted MinGWAS tightly followed the uniform distribution (Fig. 4b, c). This means that the comparison among the two approaches is very strict under the same $\alpha$. Therefore, the traditional genome-wide significance threshold of ($5 \times 10^{-8}$) used in a standard GWAS corresponds to our study-wide significance threshold. Note that when applying the threshold of $5 \times 10^{-8}$ in the adjusted MinGWAS, which corresponded to an unadjusted *p*-value of $7.52 \times 10^{-10}$, i.e., the same number of SNPs survived $7.52 \times 10^{-10}$ before and $5 \times 10^{-8}$ after the *getCoef* calibration, is in fact 1.6 folds more stringent than the study-wide significance threshold of $1.2 \times 10^{-9}$ in the previous MinGWAS. This observation is consistent with our simulation scenario 9 showing that adjusting for MinGWAS using the estimated number of independent traits is prone to inflation under stringent thresholds. Below, MinGWAS refers to our *getCoef* calibrated MinGWAS unless otherwise specified.

Under the study-wide significance threshold ($5 \times 10^{-8}$), C-GWAS identified 1200 SNPs from 23 distinct loci, while MinGWAS only identified 155 SNPs from 6 distinct loci. Not unexpected, the 6 loci identified by MinGWAS were the same 6 loci previously identified in the original study of Xiong et al.[4]. Moreover, they were included in the 23 loci identified by C-GWAS. Among the 17 loci exclusively identified by C-GWAS, 3 (18%) were novel and 14 (82%) were reported in previous single-trait face GWASs[5,9–21] including the study of Xiong et al.[4].

Moreover, under the study-wide suggestively significant threshold estimated for C-GWAS at $3 \times 10^{-6}$, C-GWAS identified a total of 2591 SNPs from 56 distinct loci compared to only 418 SNPs from 17 distinct loci identified by MinGWAS (Fig. 4d–f), of which the majority (13)

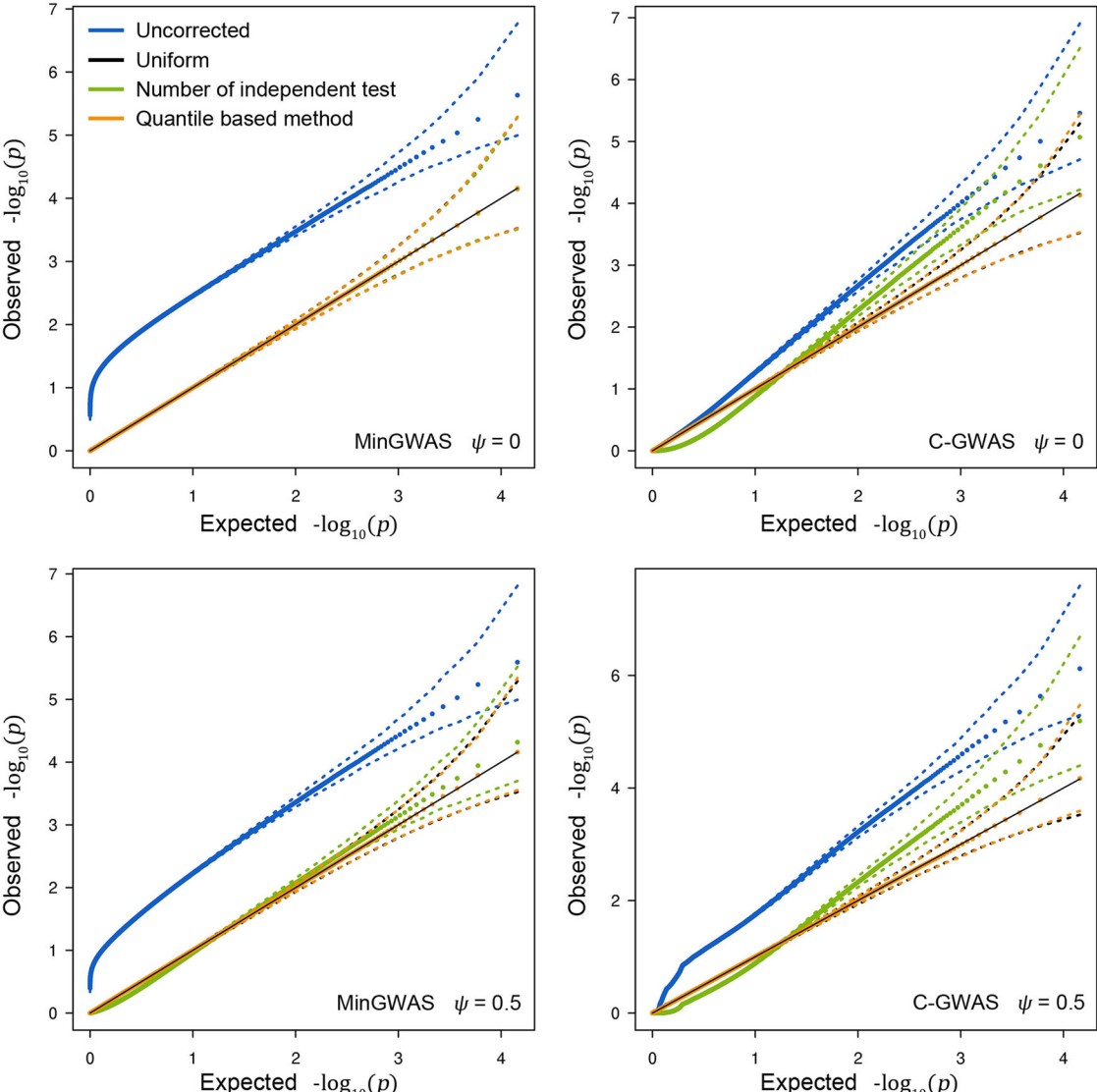

**Fig. 3 | The null distributions before and after calibration from C-GWAS and MinGWAS via simulations.** Vectors of $T$ statistics and corresponding two-side $p$-values of 30 GWASs were simulated under the null, each consisting of 10,000 SNPs without any true effect. All pairs of $T$ statistics vectors were simulated to have the same $\psi$. C-GWAS and MinGWAS were used to combine the 30 simulated GWASs. Null distributions of C-GWAS (the right column) and the MinGWAS (the left column) were obtained at $\psi = 0$ (the upper row) and $\psi = 0.5$ (the lower row). The crude $p$-values (blue), the $p$-values after the Tippett's method based correction of the number of independent tests obtained from the *getNtest* function (green), the $p$-values calibrated using the *getCoef* function (orange), and the uniform distribution (black) are distinguished in different colors. The median and 95% intervals at all quantiles are indicated using dots and dashed lines respectively.

overlapped with C-GWAS findings. Similar to the study-wide significant threshold of $5 \times 10^{-8}$, the threshold of $3 \times 10^{-6}$ in our MinGWAS is 1.15 folds more stringent than the study-wide suggestive significance threshold of $5 \times 10^{-8}$ in the study of Xiong et al.[4]. The 13 loci overlapping between C-GWAS and MinGWAS all showed an order of magnitude enhanced significance in C-GWAS compared to MinGWAS. A prominent example is rs55674676 near 12q24.21 *TBX3* which demonstrated $p$-value enhancement from $2.58 \times 10^{-8}$ in MinGWAS to $3.11 \times 10^{-19}$ in C-GWAS. The 4 MinGWAS-only identified loci were all reported in previous GWASs. Of the 43 C-GWAS-only identified loci, 17 (40%) were novel and 26 (60%) were reported in previous GWASs[5,9–21] including the study of Xiong et al.[4].

We double-checked our results using the LD-score regression method[22], which estimated the LD-score intercepts close to 1.0 for both C-GWAS and MinGWAS. This analysis confirmed that the signals from both C-GWAS and MinGWAS cannot be explained by potential population sub-stratifications and other unknown factors. Compared with MinGWAS, C-GWAS gained 54% extra statistical power as

estimated using the increase in the mean $\chi^2$ statistic method described in Turley et al.[3].

We looked up in our C-GWAS results the 327 previously established face-associated SNPs that were identified in traditional single-SNP GWASs[5,9–21]. For these SNPs as well, C-GWAS $p$-values deviated further from the null than those from MinGWAS (Fig. 4g). The observation that C-GWAS boosted the significance of MinGWAS findings and re-identified a much larger number of previously established loci than did MinGWAS empirically confirmed a substantial gain in statistical power and strengthened the reliability of the C-GWAS-identified novel loci, in line with our simulation results.

Next, we performed replication analyses for the 56 regional lead SNPs identified by C-GWAS with study-wide suggestively significance in a total of 13,622 individuals not overlapping with Xiong et al.[4]. These replication data included the Rotterdam Study ($N$ = 1174 Dutch Europeans), 2774 individuals of European descent from two cohorts[5], and 9674 Chinese from three cohorts[6]. Combining replication evidence from all 6 cohorts revealed an overall highly significant deviation from

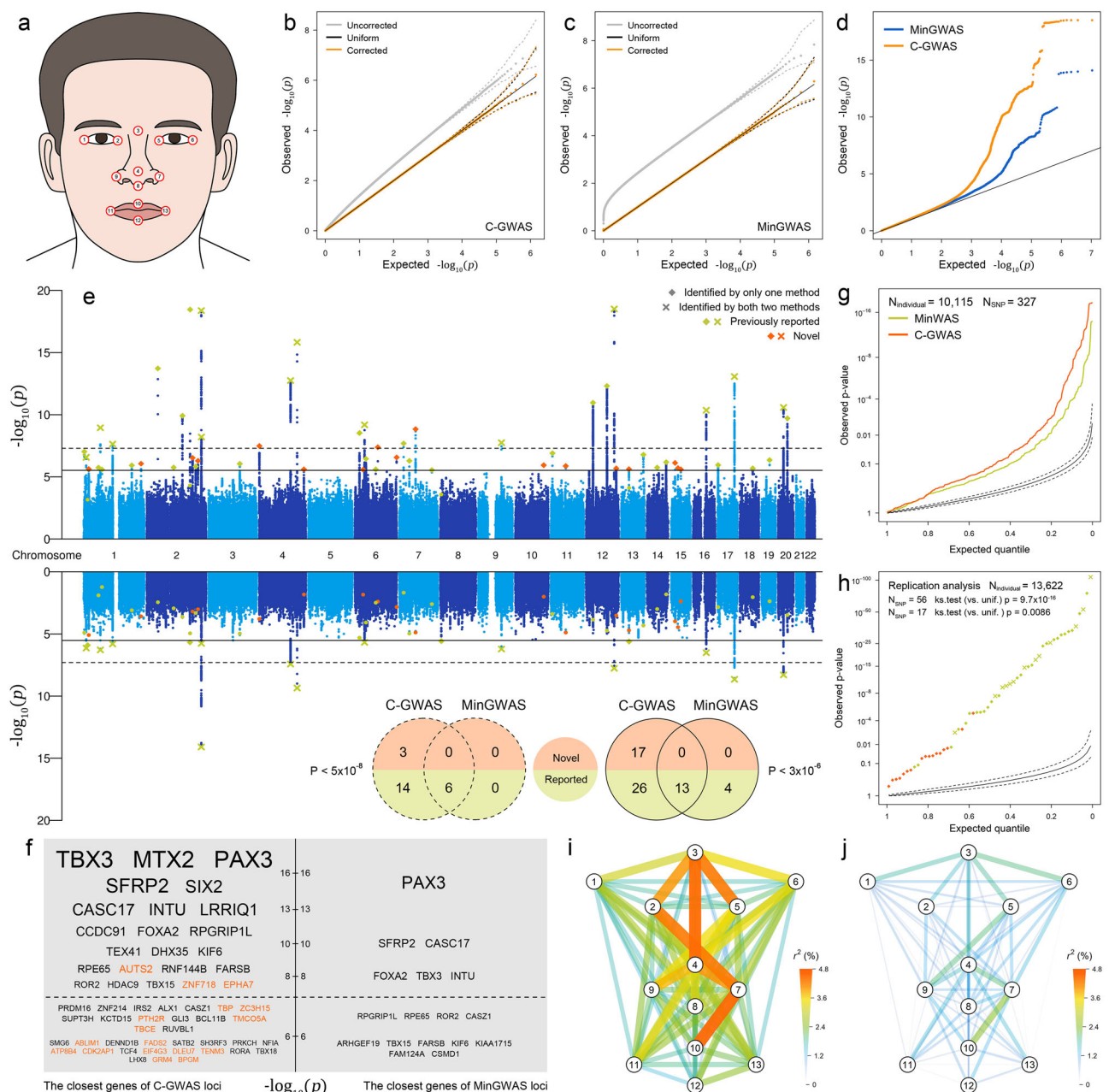

**Fig. 4 | C-GWAS analysis of 78 correlated facial traits and replication.** C-GWAS is applied to combine summary statistics of 78 GWASs based on 10,115 samples from four European cohorts[4]. **a** The 78 correlated facial traits were derived as the pairwise Euclidean distances between 13 well-defined facial landmarks. **b**, **c** A simulation analysis derived the null distributions of C-GWAS and MinGWAS in terms of crude $p$-values (gray) and $p$-values calibrated using *getCoef* (orange), which are compared with the uniform distribution (black). Median values and 95% intervals at all quantiles are indicated using dots and dashed lines. After distribution calibration, the C-GWAS results (upper part of **e**) and the MinGWAS $p$-values (lower part of **e**) are plotted using an overlayed Q–Q plot (**d**) and Miami plot (**e**). The study-wide significance threshold ($p = 5 \times 10^{-8}$) is indicated using dashed lines and the study-

wide suggestive significance threshold ($p = 3 \times 10^{-6}$) is indicated using solid lines. The overlapping between previously established and novel loci is illustrated in a Venn diagram. The closest genes to the regional lead SNPs are listed in (**f**) (orange for novel). A list of 327 previously established face-associated SNPs was looked up in the C-GWAS and MinGWAS results ($N = 10,115$, **g**). The regional lead SNPs at 56 loci from C-GWAS were looked up in the combined replication dataset ($N = 13,622$, **h**). One-side Kolmogorov-Smirnov test is used to test if the observed $p$-value distribution is deviated from the uniform distribution under the null. In the replication sample of 1174 individuals of RS, the facial variance explained by C-GWAS polygenic risk score (PRS) (**i**) is compared with that explained by MinGWAS PRS (**j**).

the expectation under the null (KS test $p = 9.7 \times 10^{-16}$, Fig. 4h). Moreover, among the 56 SNPs, 91% (51/56) showed consistent allele effects with the discovery cohort and 82% (46/56) was replicated at nominal significance ($p < 0.05$). Focusing on the 17 novel SNPs identified by C-GWAS also showed an overall significant deviation from the expectation under the null (KS test $p = 8.6 \times 10^{-3}$, Fig. 4h), with 82% (14/17) showing consistent allele effects with the discovery cohort, and 53% (9/17) replicating at the nominal significance ($p < 0.05$).

## C-GWAS outperforms MTAG and other methods in real data of the human face

Next, we compared the performance of C-GWAS and MTAG in the real face data. To this end, because MTAG could not complete the analysis of all 78 GWASs within our computational capacity, we focused on a subset of 30 GWASs largely covering the left side of the strongly symmetrical human face. Although the H0 of C-GWAS and MTAG are different, a fair comparison was achieved by adjusting the minimal $p$-

value of MTAG using the same *getCoef* function as in C-GWAS. In addition, we compared C-GWAS with four competing statistics, including mixAda, mixFisher and mixTippett proposed by Liu et al.[23] and $S_{Het}$ proposed by Zhu et al.[2] (Supplementary Table 1). These four statistics test the same H0 as C-GWAS, but do not distinguish $\Pi$ and $\Psi$ as C-GWAS and MTAG. Both C-GWAS and $S_{Het}$ consider GWAS subsets whereas others do not.

Under the study-wide significance threshold of $5 \times 10^{-8}$, C-GWAS identified 649 SNPs from 16 distinct loci, all (100%) of which represent previously established facial loci. MTAG only identified five SNPs from two loci, one (50%) of which has been previously established (Supplementary Fig. 12). The mixAda appeared the best performing among the three statistics proposed by Liu et al.[23], which identified 18 SNPs from 12 loci, one (8%) of which has been previously established. The $S_{Het}$ identified 252 SNPs from 16 loci, 12 (75%) of which overlapped with previously established loci. The 2q31.1 *MTX2* was the only locus overlapping between all methods.

Considering the study-wide suggestive threshold of $1.35 \times 10^{-6}$, C-GWAS identified 1,043 SNPs from 32 distinct loci, 22 (69%) of which represent previously established facial loci. MTAG only identified 55 SNPs from 15 loci, 3 (20%) of which has been previously established (Supplementary Fig. 12). The mixAda again was the best performing statistic among those proposed by Liu et al.[23], which identified 160 SNPs from 61 loci, 9 (15%) of which have been previously established. The $S_{Het}$ identified 880 SNPs from 56 loci, 18 (32%) of which overlapped with previously established loci. The 2q31.1 *MTX2* and 12q24.21 *TBX3* were the only loci overlapping between all methods.

The LD-score intercept of C-GWAS was very close to 1.0 as expected, whereas the MTAG showed severe deflation (mean of 30 LD-score intercepts = 0.83; sd = 0.04). In contrary, the $S_{Het}$ showed severe inflation (1.07) and the three statistics proposed by Liu et al. also showed non-negligible inflation (>1.04) (Supplementary Table 1).

Finally, we looked up the 327 previously established face-associated SNPs and it was obvious that C-GWAS *p*-values deviated much further from the null than did MTAG and mixAda *p*-values, the latter two were largely non-significant (Supplementary Fig. 12). C-GWAS replicated considerably more SNPs (49) than did $S_{Het}$ (33) after Bonferroni correction ($p = 2 \times 10^{-4}$), although $S_{Het}$ *p*-values also significantly deviated from the null (Supplementary Fig. 13).

The observation that MTAG showed severe deflations and re-identified only a small number of previously established loci appears surprising and suggests that the GWAS dataset of Xiong et al. is not suitable for MTAG. Note that the extra multiple testing burden posed to MTAG by forcing it to test our H0 had negligible impact given that MTAG performed similarly poor even without adjusting for the 30 facial traits. Complex patterns of SNP effects across different facial traits likely led to violation of the assumption in MTAG. The fact that $S_{Het}$ outperformed MTAG and mixAda indicates that considering GWAS subsets is important when the number of GWASs is large.

## C-GWAS outcome increases proportion of facial phenotype variance explained

Next, we compared C-GWAS and MinGWAS in their outcomes to genetically explain the facial phenotype variance in RS based on a polygenic risk score (PRS) analysis. The PRS consisting of the 57 suggestively significant SNPs from C-GWAS (56 regional lead SNPs plus one in low LD) and explained on average 2.28% and up to 4.51% sex- and age-adjusted facial variance (Fig. 4i). In contrast, the PRS based on MinGWAS consisting of 17 suggestively significant lead SNPs only explained on average 0.84% and up to 2.37% of facial variance (Fig. 4j). The PRS based on MinGWAS consisting of 57 top-ranked SNPs explained on average 2.03% and up to 4.33% sex- and age-adjusted facial variance (Supplementary Fig. 14), but is significantly lower than the PRS based on 57-SNP suggestively significant SNPs from C-GWAS (two-side paired Wilcoxon rank-sum test $p = 9 \times 10^{-4}$).

We then compared face PRSs between Europeans (EUR), Sub-Saharan Africans (AFR) and East Asians (EAS) in totally 1668 samples from the 1000-Genomes Project. Although the face PRSs were solely derived from Europeans, when applied to non-European populations, the C-GWAS PRS largely assembled the facial features in major continental groups[6,24,25], which appeared more consistent with anthropological knowledge than the MinGWAS PRS (Supplementary Fig. 14). For example, genetically, AFR mouth appeared larger and AFR face appeared wider than those of EUR and EAS, EUR mouth appeared smaller and EUR nose appeared narrower and more protuberant than those of AFR and EAS, and EAS face was overall less distinct, largely in the middle between AFR and EUR. Overall, these trends appeared more obvious in C-GWAS PRS than MinGWAS PRS (Supplementary Fig. 14).

## Multi-trait facial effects of C-GWAS outcome

To empirically test whether C-GWAS indeed identified loci with multi-trait effects, as expected based on the method design, we developed a statistic to empirically estimate multi-trait effect (MTE, see "Methods"). We then estimate MTE of the 56 study-wide suggestively significant SNPs identified by C-GWAS, including 13 suggestively significant also in MinGWAS. Overall, 26 eigen vectors were derived to represent the 78 GWASs after removing background correlations (see "Methods"). The 43 C-GWAS-only identified SNPs were more enriched with larger MTE (>0) values than the 13 significant SNPs identified by both C-GWAS and MinGWAS (two-side Fisher's exact test $p = 0.026$, odds ratio = 5.06). In addition, the 17 SNPs from novel loci identified with C-GWAS were significantly more enriched with larger MTE (>0) values than the 39 re-discovered SNPs reported in previous GWASs (two-side Fisher's exact test $p = 0.025$, odds ratio = 8.4, Fig. 5a).

Eight examples were examined in more detail, including two novel loci, *EIF4G3* and *CDK2AP1* with suggestively significant association and large MTE as well as six re-identified previously established loci with study-wide significant association and varying MTE: *PAX3*, *SFRP2* (large MTE), *TBX3*, *SIX2* (moderate MTE), *INTU* and *RPE65* (small MTE, Fig. 5b). Their allelic effects of the regional lead SNPs on the 26 eigen vectors displayed two clear patterns. First, the further a SNP effect deviated from the null, the more significant its association was in C-GWAS (example, *PAX3* vs *EIF4G3*, both with large MTE). Second, the earlier a SNP effect deviated from the null, the larger its MTE value was (example, *PAX3* vs *TBX3* vs *INTU*, all established loci with study-wide significant association). Consistently, SNPs with large MTE values were generally associated with a larger number of facial traits with different effects, which was seen not only for the examples mentioned here (Fig. 5c), but also for other face-associated SNPs from C-GWAS (Supplementary Fig. 15). These results empirically demonstrate the multi-trait effects of C-GWAS findings and illustrate the power of C-GWAS in identifying genetic variants with various degrees of multi-trait effects.

## Biological annotations of C-GWAS and MinGWAS findings

Gene ontology analyses were conducted for comparing the C-GWAS (83 genes in 56 loci) and MinGWAS (24 genes in 17 loci) findings in Gene Ontology (GO), Human Phenotype (HP), and Mouse Phenotype (MP) databases. Overall, C-GWAS loci harbored genes significantly enriched with a larger number of facial morphogenesis relevant terms relative to MinGWAS loci (Fig. 6a), with the term "abnormality of external nose" being the most significant in HP ($p = 3.36 \times 10^{-9}$). For the significant biological terms found by both C-GWAS and MinGWAS, those from C-GWAS were generally more significant than those from MinGWAS. The vast majority of terms from MinGWAS-identified loci were covered by terms from C-GWAS-identified loci. The unique terms from C-GWAS-identified loci not only revealed the facial morphogenesis pathway, but also revealed the nervous development pathway including 'central nervous system development' and 'nervous differentiation', which is consistent with the recent observation of

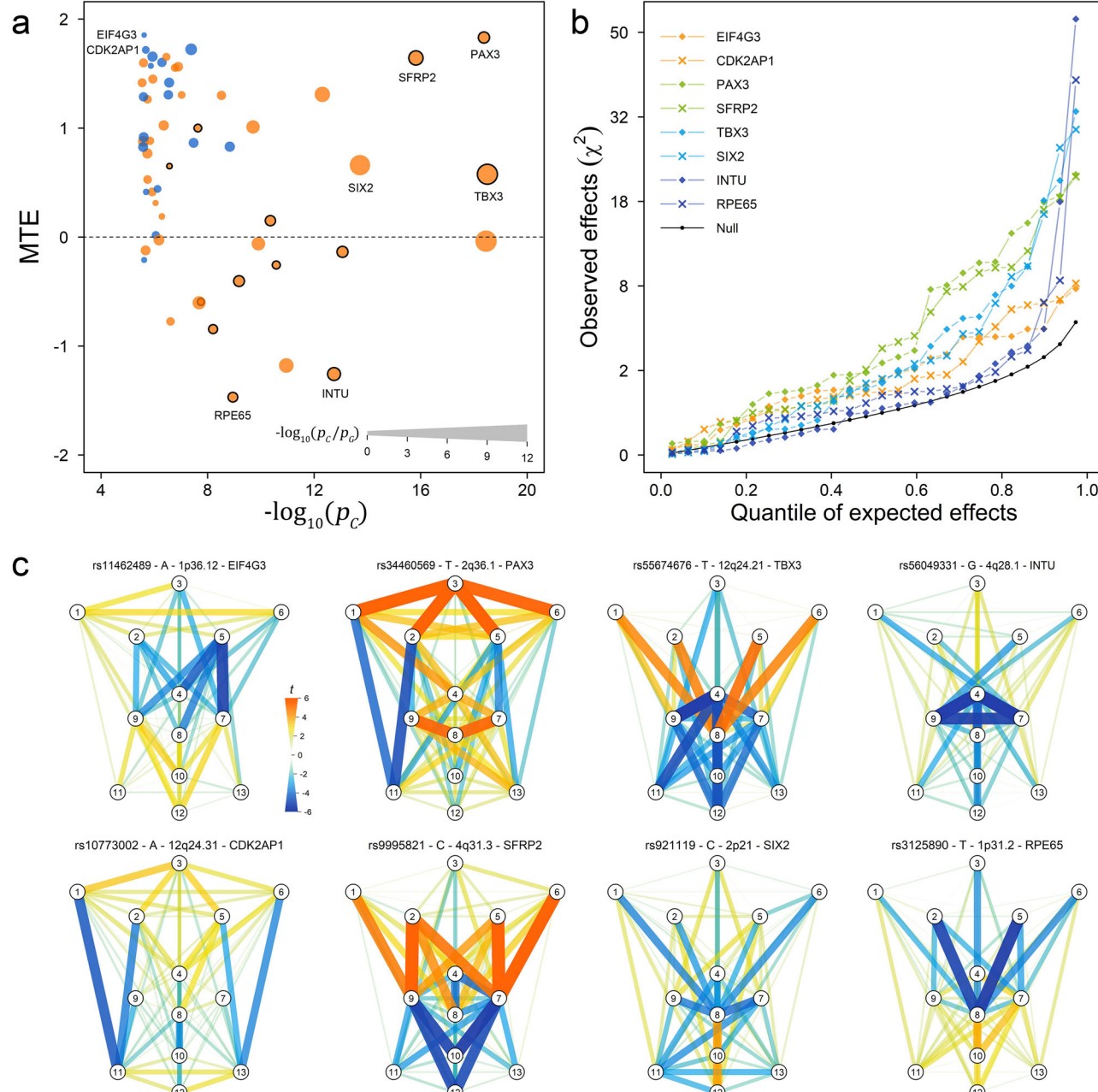

**Fig. 5 | Multi-trait effects of the face-associated SNPs with examples.** The estimated multi-trait effect (MTE, *y*-axis) of the 57 C-GWAS-identified study-wide suggestively significant regional lead SNPs were plotted against their $-\log_{10}(p)$ from C-GWAS ($-\log_{10}(p_C)$, *x*-axis, **a**). SNPs are indexed according to their closest genes for illustrative purpose. SNPs are distinguished with two colors of orange for previous known face-associated loci and blue for novel loci. The boost of significance of C-GWAS in relative to MinGWAS ($-\log_{10}(p_C/p_G)$) is indicated using the dot size. The SNPs that are suggestively significant in both C-GWAS and GWAS are indicated using circles with solid lines. The squared projected effects ($\chi^2$) are plotted against the quantile of the expected effects on the 26 projected vectors (see "Methods" for details, **b**). The expected effects under the null are indicated using black dots connected by black lines. The observed effects are detailed for eight examples with large or small values of MTE, including *EIF4G3*, *CDK2AP1*, *PAX3*, *SFRP2*, *TBX3*, *SIX2*, *INTU*, and *RPE65*, which are distinguished using different colors and dot shapes. The effect of the lead SNPs in terms of the T-statistic from GWASs on 78 facial traits are superimposed on face maps (**c**). Face maps of full 57 C-GWAS lead SNPs are available in Supplementary Fig. 15.

overlapping findings between GWASs of facial traits and those of brain traits[26].

Cranial neural crest cells (CNCC) reflect a migratory cell population at early human development involving peripheral nervous system, skin pigment cells, and craniofacial bones. Previously, CNCC regulatory activity in face-associated loci has been used to elucidate their functional roles in facial morphogenesis[4,5,17]. Here, we applied the recently published regulatory network of CNCC[27] to annotate our C-GWAS and MinGWAS findings. Among 56 C-GWAS-identified loci, 26

(46%) were found to contain CNCC-associated regulatory elements (REs), while only 5 (29%) out of 17 MinGWAS-identified loci contained CNCC-associated REs (ISTAT[28] enrichment $p = 2.16 \times 10^{-4}$ for C-GWAS and $p = 0.04$ for MinGWAS). Furthermore, the subnetwork obtained by C-GWAS-identified loci regulates more target genes (TGs) than the one obtained by MinGWAS-identified loci (54 vs. 8 TGs, Fig. 6b, c). For example, ALX1 and TCF4, which are involved in CNCC development[29] and identified in regulatory network of CNCC[27] as the core transcription factors[30], are exclusively from C-GWAS. These results

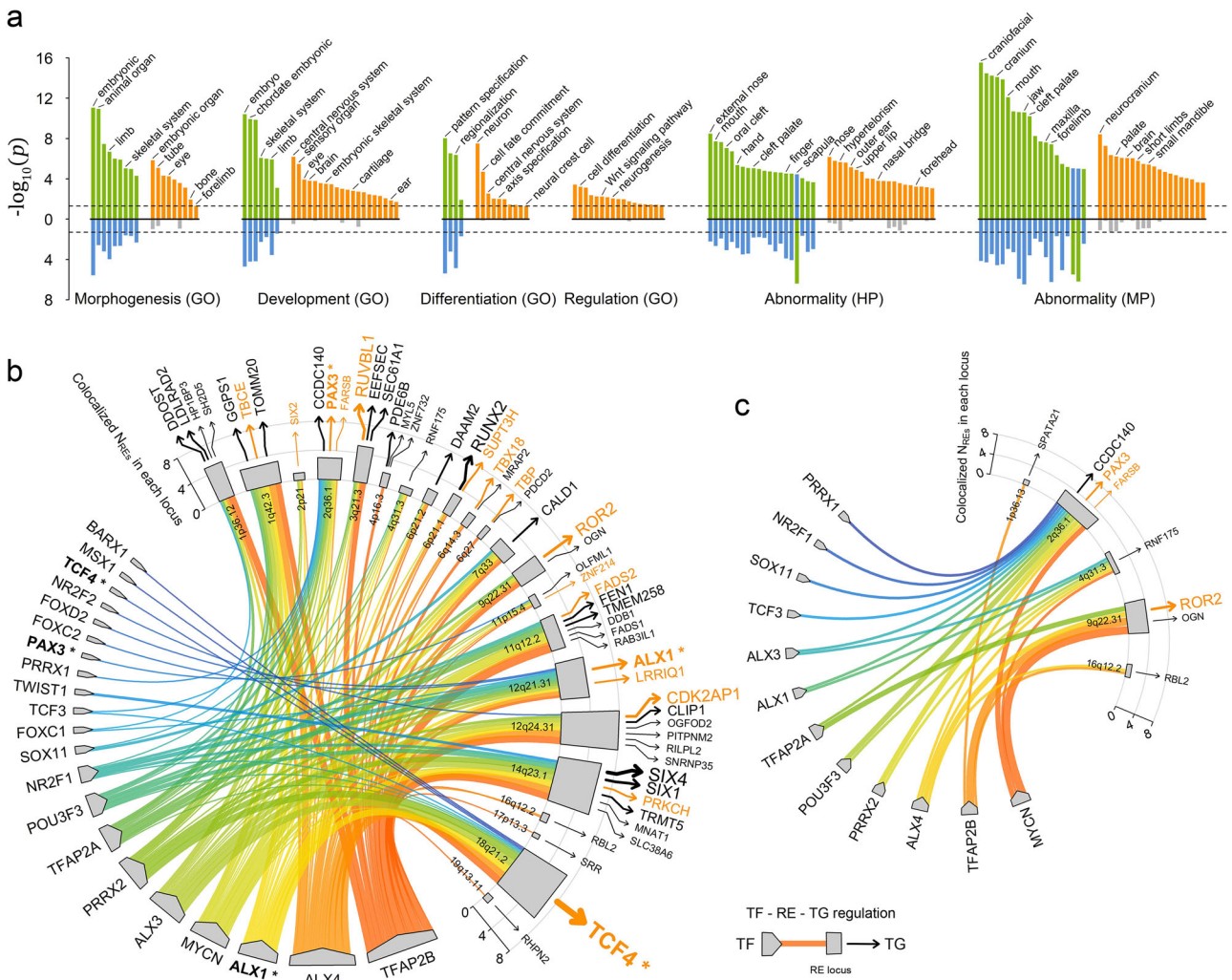

**Fig. 6 | Biological annotation of significant findings. a** Gene ontology (GO), human phenotype ontology (HP), mouse phenotype ontology (MP) terms enrichment analyses were carried out based on C-GWAS (83 genes in 56 loci) and Min-GWAS (24 gene in 17 loci) findings. Biological terms with their Bonferroni corrected enriched $p$-values from C-GWAS and MinGWAS were plotted above and below the black solid line, respectively. Shared and significant terms are indicated in green (more significant) or blue (less significant). Unique terms are indicated in orange with those non-significant in gray. The dashed lines indicate Bonferroni corrected

$p = 0.05$. A regulation network in CNCC is derived based on C-GWAS finding (**b**, 56 loci) and MinGWAS finding (**c**, 17 loci). Transcription factor (TF)−regulation element (RE)−target gene (TG) regulation triplets are superimposed on chord diagrams and are distinguished using colors. The TF-RE edge width represents the number of REs associated with TF. TGs linked to their RE are highlighted and the number of RE-TG is proportional to arrows size. Closest genes to lead SNPs are highlighted in orange text. The genes that are both TF and TG are marked using "*".

demonstrated that C-GWAS loci were more biologically interpretable than MinGWAS loci.

Finally, we investigated the colocalization of genetic association at 56 C-GWAS-identified loci with eQTL signals by using expression data from 22 classified tissues potentially relevant to facial morphogenesis. Previously, a smaller scaled colocalization analysis had been conducted for 20 face-associated loci in 6 tissues[31], which highlighted *EEFSEC* with evidence of colocalization. Our study confirmed this finding and additionally revealed many novel loci with evidence of colocalization in various tissues. Overall, 38 genes in 17 loci were identified with large posterior probabilities (PP4>0.7) of colocalization in at least one tissue (Fig. 7a). These included 23 genes in 7 novel loci identified by C-GWAS, i.e., 1q42.3, 2q32.1, 2q34, 4p16.3, 11q12.2, 12q24.31, and 13q14.3. Two novel (*RP11-410E4.1* at 2q32.1 and *DLEU7-AS1* at 13q14.3) and two previously established face-associated genes (*RPGRIP1L* at 16q12.2 and *KCTD15* at 19q13.11, Fig. 7b−e) colocalized with eQTLs in a large number of different tissues (≥9). Thirteen out of the 17 loci contained at least one colocalized genes that were different to the genes closest to the regional lead SNPs. These results provided

direct evidence for functional involvement of some of the face-associated SNPs, proposed a different set of functional candidate genes, and strengthened the reliability of 7 novel loci from C-GWAS.

### Novel loci highlighted by C-GWAS
Among the 17 novel loci identified by C-GWAS, 13 are worth highlighting as they were supported by the replication study, the CNCC network analysis, or the colocalization analysis (Supplementary Data 1), of which we mention the most prominent locus here and the other 12 in Supplementary Note 1. The strongest supportive evidence was observed at the novel locus 12q24.31, where the lead SNP rs10773002 is an intron variant of *CDK2AP1*. The A-allele of rs10773002, minor in Europeans ($f = 0.24$) and Asians ($f = 0.25$) but major in Africans ($f = 0.79$), was nominally significantly associated with a wide spread of facial phenotypes, mainly contributing to reduced facial width and increased facial length (Fig. 5c). The multi-trait effect of this SNP was successfully replicated ($p = 6.9 \times 10^{-4}$). The REs in the vicinity of the face-associated SNPs at this locus regulate several nearby genes, including *CDK2AP1* in the CNCC subnetwork.

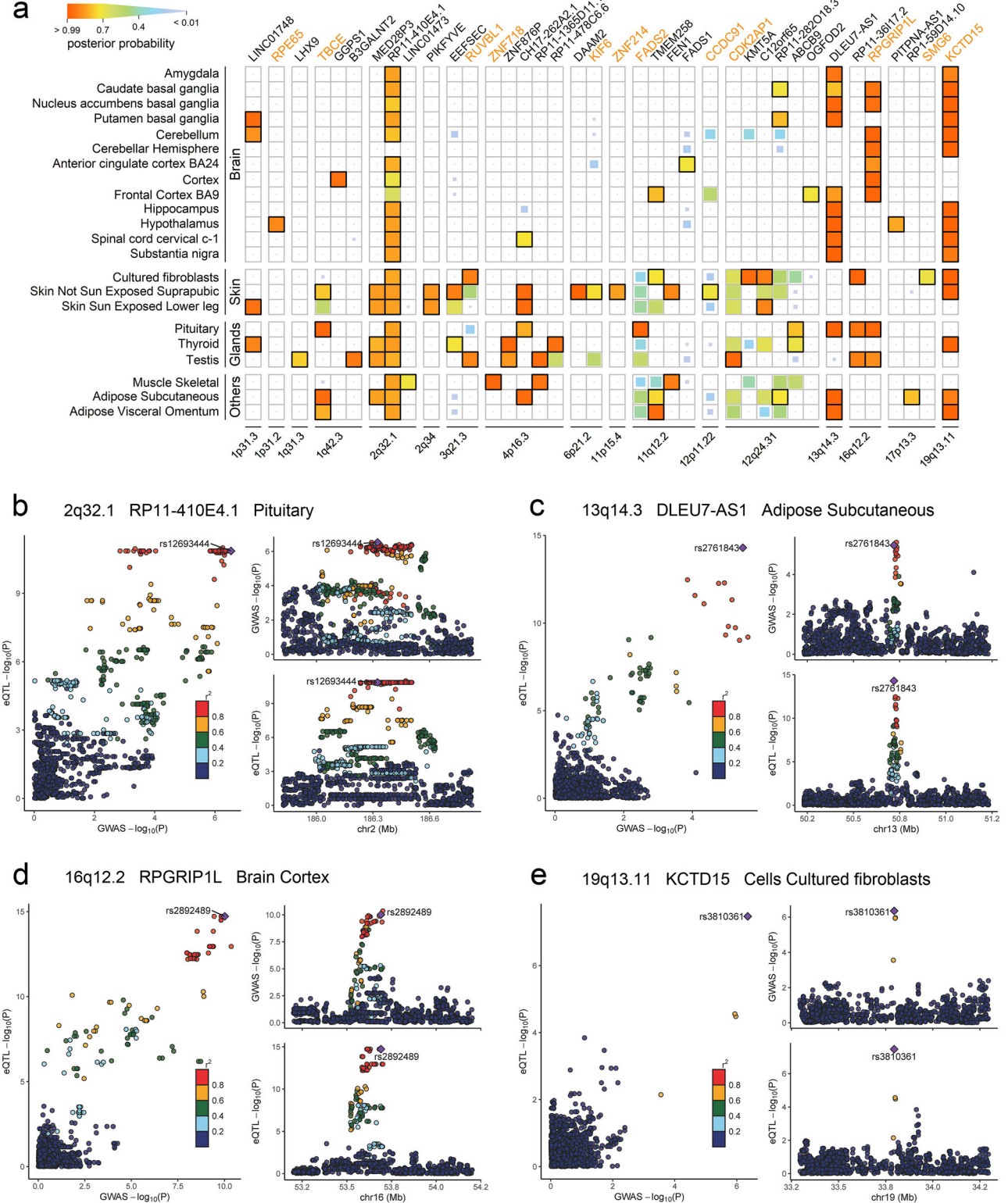

**Fig. 7 | Colocalization analysis of C-GWAS significant findings.** Colocalization analysis was carried out for the 56 loci from C-GWAS in 22 tissues from GTEx. **a** Posterior probabilities in the heatmap are indicated using the box size and the gradience in color (posterior probabilities > 0.7 in solid boxes). Closest genes to the regional lead SNPs are indicated using orange text. Four examples of eQTL

colocalizations at distinct loci in different tissues (**b**–**e**), where the regional lead SNPs were cis-associated with gene expressions (lower right) and were simultaneously associated with facial variation in C-GWAS (upper right), suggesting that these SNPs are responsible for both the cis-eQTL signal and the C-GWAS signal (left).

Face-associated SNPs in this locus colocalized with eQTLs of *CDK2AP1* and nearby genes in testis (PP4 = 0.97) and other tissues. The protein encoded by *CDK2AP1* is a cyclin-dependent kinase 2-associated protein, which plays a role in cell-cycle, embryonic stem cell differentiation and epigenetic regulation. SNPs in this gene have been associated with educational attainment and waist-to-hip ratio in previous GWAS[32,33].

## Discussion

In this study, we introduce C-GWAS, a method to integrate GWAS results of multiple correlated traits with an increased statistical power in detecting multi-trait effects. EbICoW is originated from the inverse variance weighting method with three extensions, which allows using the joint distributions of both null and true effects to detect multi-trait effects. This is particularly powerful when $\pi$ can be clearly differentiated from $\psi$, e.g., when the genetic determinants shared between traits are substantial and/or when samples do not overlap or partially overlap. When it is difficult to distinguish $\pi$ from $\psi$, Wald, which does not rely on the joint distribution of true effects, is more powerful, e.g., when traits are mainly determined by their own unique genetic factors and/or when samples overlap completely or mostly. Utilizing the statistically complementary properties of EbICoW and Wald, C-GWAS provides excellent power in a variety of complex situations, as we have demonstrated via simulations and in real face data.

In C-GWAS, the decision on whether or not using EbICoW to combine a given pair of GWASs in each iteration is made via a self-adaptive data-driven approach. Such decisions, optimal regarding data fitness, may not always fit the true underlying genetic architectures of the studied traits. This may affect the power in detecting multi-trait effects, but not the *p*-value distributions under the null, as double guaranteed by both the statistical nature of EbICoW and our approach for distribution calibration.

Coming with C-GWAS, three methods were additionally developed for inflation estimation, background correlation estimation, and distribution calibration, respectively. The first two methods provide accurate estimates of inflation and background correlation, which is critical for C-GWAS to work properly. Compared with previous LD score-based methods[22,34], our methods abandon LD-related calculation for computational efficiency, while maintaining similarly levels of accuracy and estimation error. More importantly, our methods have less limitations in the presence of GWASs with different genomic backgrounds. The third method for distribution calibration uses a quantile-based function that can convert any given distribution to a uniform distribution, which guarantees unbiased estimates as long as the null can be obtained via simulations, which allows fair comparisons between C-GWAS, MinGWAS, and any given standard GWAS. These methods are generic and flexible (requiring only summary statistics), so that they can be widely applied to genetic studies of complex traits.

C-GWAS has the flexibility to be conducted as a whole or by parts considering only EbICoW or TWT. We recommend to carry out C-GWAS as a whole when the expected differences between $\Pi$ and $\Psi$ among a large number of GWASs are largely unknown, so that our self-adaptive decision-making algorithm can assist in optimizing statistical power. In cases of non-overlapping GWASs where all traits are expectedly highly correlated, we recommend using EbICoW only. Generalization of C-GWAS to other omics data requires the null following multidimensional standard normal distributions.

## Methods
### Ethnic statement
This study complies with all relevant ethical regulations. The Rotterdam Study has been approved by the Medical Ethics Committee of the Erasmus MC (registration number MEC 02.1015) and by the Dutch Ministry of Health, Welfare and Sport (Population Screening Act WBO, license number 1071272–159521 PG). The Rotterdam Study has been entered into the Netherlands National Trial Register (NTR; www.trialregister.nl) and into the WHO International Clinical Trials Registry Platform (ICTRP; www.who.int/ictrp/network/primary/en/) and under shared catalog number NTR6831. All participants provided written informed consent to participate in the study and to have their information obtained from treating physicians. Other data used here come from previous publications[4–6], where the respective ethics statements are provided.

### Definition of background and effect correlations
Consider $K$ GWAS of $M$ SNPs, under the null, for each SNP, its $T$ statistics from different GWASs follow a sum of two multidimensional normal distributions including the estimate error and the inflation as $\mathbf{t} = (t_1, t_2, \ldots, t_K)^\mathrm{T}$. The covariance matrix of the estimate error has a diagonal of 1 and an off-diagonal of zero, which may deviate from zero in presence of sample overlap and non-genetic phenotype correlations ($\rho$), e.g., caused by shared environmental or unknown factors. The mean squared inflation is zero and may deviate from zero in the presence of cryptic relatedness and population stratification. In the absence of inflation or when inflation is well controlled, the correlation $\psi$ ($K = 2$) or the correlation matrix $\Psi$ ($K > 2$) between the summary statistics of GWAS $T$ statistics, $\mathbf{T} \in (\mathbf{T}_1, \mathbf{T}_2, \ldots, \mathbf{T}_K)$, is defined as 'background correlation'. For two GWASs of sizes $N_1$ and $N_2$ with an overlapping sample of size $N_c$, ideally $\psi = \frac{N_c}{\sqrt{N_1 N_2}} \rho$. Therefore, when the sample does not overlap at all or in the absence of non-genetic phenotype correlations, $\Psi$ is an identity matrix.

Under the polygenicity, a proportion of SNPs additionally has true effects in at least one GWAS. Similar to $\psi$ or $\Psi$, the correlation $\pi$ ($K = 2$) or the correlation matrix $\Pi$ ($K > 2$) between the true effects of $\mathbf{T}$ is defined as 'effect correlation'. With these definitions, the relationship between any given numbers of GWASs can be described using $\Psi$ and $\Pi$.

### C-GWAS flowchart
The C-GWAS flowchart is shown in Supplementary Fig. 1. Consider $K$ GWASs $\mathbf{G} \in (\mathbf{G}_1, \mathbf{G}_2, \ldots, \mathbf{G}_K)$, where $\mathbf{G}_i$ contains effect sizes, standard errors, and $T$ statistics of the GWAS $i$, C-GWAS takes $\mathbf{G}$ as the input, which is passed to i-EbICoW followed by TWT and distribution calibration. The important parameters for i-EbICoW include the vector of inflation factors $\mathbf{s} = (s_1, s_2, \ldots, s_K)^\mathrm{T}$ for all GWASs, estimated using the *getI* function; the background correlation matrix $\Psi$, estimated using the *getPsi* function; the effect correlation matrix $\Pi$, estimated using the *getPi* function, and the effect vector $\mathbf{h}$, estimated using the *getH* function. i-EbICoW iteratively combines GWASs in pairs, and the order of combination is determined by the matrix $\mathbf{D}$, which is the difference between $\Pi$ and $\Psi$. The core functions of i-EbICoW involve *EbICoW*, *optimize*, and *evaluate*. *EbICoW* combines two GWAS based on an effect-based inversed covariance weighting method. *Optimize* optimizes the result of *EbICoW* within three possible choices of $\mathbf{h}$. *Evaluate* makes the decision whether or not accepting the result from *Optimize* by comparing the resultant *p*-values with those from Wald and Min-GWAS. $\mathbf{D}$ is updated in each iteration until all elements of $\mathbf{D}$ are sufficiently small. In such a way, i-EbICoW generates a set of new $\mathbf{G}$ so that no result from *Optimize* for any pair of $\mathbf{G}$ are accepted by *Evaluate*. These newly derived $\mathbf{G}$ are further passed to TWT and distribution calibration. TWT analyzes i-EbICoW results in a SNP-specific manner. For each SNP, TWT takes the subset of $\mathbf{G}$ with the minimal *p*-value from i-EbICoW as the initial subset, then extends to a series of subsets under a gradient ($n = Q$) of preset *p*-value thresholds $\mathbf{r} = (r_1, r_2, \ldots, r_Q)$, and then conducts the Wald test on all subsets to obtain the combined *p*-values of all subsets. The analysis for each SNP under $Q$ thresholds is realized through a nested loop. The distribution calibration is achieved in an empirical manner via simulations through two progressive steps. The first step is to correct all intermediate results to make them comparable with each other. It uses *getNtest* to estimate the number of

independent tests from a larger number of dependent hypothesis testing results via simulations, then uses the *Tippett* function to correct the *p*-value for each TWT combinations using the number of independent tests. The second step is to guarantee the final results from C-GWAS under the null follow the uniform distribution. It uses *getCoef* to build a correction model by fitting the observed distribution under the null (via simulations) with the uniform distribution for all quantiles. Then the minimal *p*-value of all results from the first step is further corrected based on the quantile specific coefficients from *getCoef*. In this way, C-GWAS produces the final results as a single vector of combined *p*-values, which follows the uniform distribution under the null, so directly comparable with the *p*-values from any standard GWAS.

### Effect-based inversed covariance weighting (EbICoW)

We propose EbICoW to allow joint analysis of multiple correlated phenotypes with overlapping samples. Let $\boldsymbol{\beta}$ be the vector of effect sizes from $K$ GWASs for a SNP, $\boldsymbol{\beta} = (\beta_1, \beta_2, \ldots, \beta_K)^T$. The null of EbICoW is the absence of any effect i.e., $H_0$: $\boldsymbol{\beta} = 0$, and the alternative is that at least one element of the effect vector deviates from 0. Below, we formulate EbICoW and show that the test statistics from EbICoW asymptotically follow the standard normal distribution. We prove that the traditional Inversed Variance Weighting (IVW) based meta-analysis is a special case of EbICoW when the off-diagonal of $\boldsymbol{\Psi}$ is 0 and the off-diagonal of $\boldsymbol{\Pi}$ is 1. We also prove that the $S_{Hom}$ proposed by Zhu et al.[2] is a special case of EbICoW when the true effect is the same for all GWAS and the off-diagonal of $\boldsymbol{\Pi}$ is 1.

For each SNP, the traditional IVW based meta-analysis combines results from $K$ non-overlapping GWASs of the same phenotype, requiring that the off-diagonal of $\boldsymbol{\Psi}$ is 0 and the off-diagonal of $\boldsymbol{\Pi}$ is 1. The combined effect $\beta_c$ from the meta-analysis represents the weighted sums of $K$ effects, $\beta_c = \mathbf{w}^T \boldsymbol{\beta}$, where $\boldsymbol{\beta} \sim N(0, \mathbf{V})$, $\mathbf{V}$ is a $K$ by $K$ diagonal matrix, and the vector of weights $\mathbf{w} = (w_1, w_2, \ldots, w_K)^T$ is inversely proportional with the variance of $\boldsymbol{\beta}$, i.e., diagonal elements of $\mathbf{V}$, and the sum of elements in $\mathbf{w}^T$ is set to 1. In practice, $\mathbf{w}^T$ can be constructed using the vector of reciprocal standard errors $\mathbf{v} = (1/\sigma_1, 1/\sigma_2, \ldots, 1/\sigma_K)$ or the samples sizes $\mathbf{n} = (n_1, n_2, \ldots, n_K)$ from $K$ GWASs. Let $\mathbf{b}$ be a vector of 1 with length $K$,

$$\mathbf{w}^T = \frac{\mathbf{v} \circ \mathbf{v}}{\mathbf{b}^T (\mathbf{v} \circ \mathbf{v})^T} \quad \text{or} \quad \mathbf{w}^T = \frac{\mathbf{n}}{\mathbf{b}^T \mathbf{n}^T} \quad (4)$$

For each SNP, the test statistics of IVW ($t_V$) can be described using $\boldsymbol{\beta}$ and $\mathbf{v}$ or using $\mathbf{t}$ and $\mathbf{n}$ as

$$t_V = \frac{\beta_c}{\sqrt{\mathrm{Var}(\beta_c)}} = \frac{\mathbf{v} \circ \mathbf{v}}{\sqrt{\mathbf{b}^T (\mathbf{v} \circ \mathbf{v})^T}} \boldsymbol{\beta} \quad \text{or}$$

$$t_V = \frac{\mathbf{v}}{\sqrt{\mathbf{b}^T (\mathbf{v} \circ \mathbf{v})^T}} \mathbf{t} = \frac{(\sqrt{n_1}, \sqrt{n_2}, \ldots, \sqrt{n_K})}{\sqrt{\mathbf{b}^T \mathbf{n}^T}} \mathbf{t}$$

Under the null, $t_V$ asymptotically follows the standard normal distribution.

When the off-diagonal elements of $\boldsymbol{\Psi}$ is not 0, $\boldsymbol{\beta} \sim N(0, \mathbf{C})$, where $\mathbf{C}$ is the covariance matrix of $\boldsymbol{\beta}$ under the null, $\mathbf{w}^T$ can be derived under the same frame of weighted sums using the Lagrange multiplier method (Supplementary Method A). Thus, we have

$$\mathbf{w}^T = \frac{\mathbf{b}^T \mathbf{C}^{-1}}{\mathbf{b}^T \mathbf{C}^{-1} \mathbf{b}} \quad (5)$$

The revised statistics $t_C$ is expressed as follows,

$$t_C = \frac{\beta_c}{\sqrt{\mathrm{Var}(\beta_c)}} = \frac{\mathbf{b}^T \mathbf{C}^{-1}}{\sqrt{\mathbf{b}^T \mathbf{C}^{-1} \mathbf{b}}} \boldsymbol{\beta} = \frac{\mathbf{e}^T \boldsymbol{\Psi}^{-1}}{\sqrt{\mathbf{e}^T \boldsymbol{\Psi}^{-1} \mathbf{e}}} \mathbf{t} \quad (6)$$

where $\mathbf{e}$ is the vector of weights only relying on $\mathbf{n}$, i.e., $\mathbf{v}^T$ or $(\sqrt{n_1}, \sqrt{n_2}, \ldots, \sqrt{n_K})^T$, but not on $\boldsymbol{\Psi}$ or $\boldsymbol{\Pi}$. Under the null, the revised $t_C$ in (6) also asymptotically follows the standard normal distribution. The $S_{Hom}$ proposed by Zhu et al. has the same null as ours thus directly comparable with ours. $S_{Hom}$ is based on a chi-squared statistic with 1 $df$, which is in fact equivalent to the $t_C^2$ in (6) when it uses $\mathbf{v}^T$ or $(\sqrt{n_1}, \sqrt{n_2}, \ldots, \sqrt{n_K})^T$ as weights. The only difference here is that the effects can be directly estimated as $\mathbf{w}^T \boldsymbol{\beta}$ using our revised weight but not available from $S_{Hom}$. Although this $t_C$ statistic can be used for joint analysis of multiple GWASs when the off-diagonal of $\boldsymbol{\Psi}$ is not 0, it has limited power when expected true effect $\boldsymbol{\delta} = (\delta_1, \delta_2, \ldots, \delta_K)^T$ is not the same for all GWAS or the off-diagonal of $\boldsymbol{\Pi}$ deviates from 1. The loss of power becomes obvious when the $K$-based weights are the same across all GWAS but the true effects are different. The loss of power also become obvious when the off-diagonal of $\boldsymbol{\Pi}$ is negative, and particularly pronounced when it is approaching -1. To overcome these limitations, we made two improvements. First, we introduce an effect vector, $\mathbf{h} = (h_1, h_2, \ldots, h_K)^T$, which is a projection of the true effect covariance matrix $\mathbf{H}$ as $\mathbf{h} = \mathbf{b}^T \mathbf{H}$. $\mathbf{H}$ is a $K$ by $K$ covariance matrix of $\boldsymbol{\delta}$ among the genome-wide SNPs, where $H_{ij} = \mathrm{E}(\delta_i \delta_j)$, $i \in \{1, 2, \ldots, K\}$ and $j \in \{1, 2, \ldots, K\}$. We use $\mathbf{h}$ to revise $\mathbf{w}^T$ as

$$\mathbf{w}^T = \frac{\mathbf{h} \mathbf{C}^{-1}}{\mathbf{h} \mathbf{C}^{-1} \mathbf{b}} \quad (7)$$

A nice feature of (7) is that $\mathbf{h}$ only affect power but not $\alpha$. In practice, $\mathbf{h}$ can be estimated using the summary statistics from GWAS (details see estimation of $\mathbf{h}$ section). It can be shown that (5) is a special case of (7) when $\boldsymbol{\delta}$ is same across all GWAS,

$$\mathbf{w}^T = \frac{\mathbf{b}^T \mathbf{H} \mathbf{C}^{-1}}{\mathbf{b}^T \mathbf{H} \mathbf{C}^{-1} \mathbf{b}} = \frac{(\mathbf{b}^T \circ \mathbf{b}^T) \mathbf{C}^{-1}}{(\mathbf{b}^T \circ \mathbf{b}^T) \mathbf{C}^{-1} \mathbf{b}} = \frac{\mathbf{b}^T \mathbf{C}^{-1}}{\mathbf{b}^T \mathbf{C}^{-1} \mathbf{b}} \quad (8)$$

It can also be empirically shown that when the $\boldsymbol{\delta}$ is different across GWASs, the test based on (7) has higher power than that based on (5) in most circumstances.

The second improvement is that we take the direction of combination into consideration, i.e., by changing the direction of combination if $\pi$ is smaller than $\psi$ while keeping $\mathbf{w}^T \mathrm{sign}(\boldsymbol{\Pi}\text{-}\boldsymbol{\Psi})_i$ equals to $\mathbf{b}^T$, where the function *sign* is defined as $\mathrm{sign}(x) = 1$ when $x \geq 0$ and $\mathrm{sign}(x) = -1$ when $x < 0$ for a variable $x$ so that $[\mathrm{sign}(\mathbf{X})]_{ij} = \mathrm{sign}(X_{ij})$ for a matrix $\mathbf{X}$. For $k \in \{1, 2, \ldots, K\}$,

$$\mathbf{w}^T = \frac{\mathbf{h} \mathbf{C}^{-1}}{\mathbf{h} \mathbf{C}^{-1} \mathrm{sign}(\boldsymbol{\Pi} - \boldsymbol{\Psi})_k}$$

where $\mathbf{h} = \mathrm{sign}(\boldsymbol{\Pi} - \boldsymbol{\Psi})_k \mathbf{H}$. In this way, the direction of elements in the effect covariance matrix is changed by comparing the effect and background correlations. It can be empirically shown that when $\pi$ is smaller than $\psi$, the test based on (8) has higher power than that based on (7) in combining two GWASs. However, (8) does not generate unique combined results for combining more than two GWASs, meaning that for obtaining one-dimensional summary statistics in the form of traditional GWAS, an iterative procedure is preferred (see Combining multiple GWASs section). Summing up, the revised $t_E$ of our proposed EbICoW is expressed as

$$t_E = \frac{\beta_c}{\sqrt{\mathrm{Var}(\beta_c)}} = \frac{\mathbf{h} \mathbf{C}^{-1}}{\sqrt{\mathbf{h} \mathbf{C}^{-1} \mathbf{h}^T}} \boldsymbol{\beta} = \frac{(\mathbf{h} \circ \mathbf{e}^T) \boldsymbol{\Psi}^{-1}}{\sqrt{(\mathbf{h} \circ \mathbf{e}^T) \boldsymbol{\Psi}^{-1} (\mathbf{h} \circ \mathbf{e}^T)^T}} \mathbf{t}$$

Note that a statistically desired feature of EbICoW is that $t_E$ asymptotically follows the standard normal distribution under the null regardless to $\mathbf{h}$.

## Wald test, Wald(G)

EbICoW uses additional information provided by effect correlation $\Pi$ and effect-based factor to improve the power of detecting SNPs with true effects. However, the increase of power depends on $|\Pi\text{-}\Psi|$. For example, considering two GWAS, the smaller $|\pi\text{-}\psi|$ is, less obvious is the power improvement of EbICoW. To overcome this limitation, we used the multi-trait chi-squared statistic proposed previously by Bolormaa et al.[1] when all elements of $|\Pi\text{-}\Psi|$ are small. The nature of the multi-trait chi-squared test is Wald's test. For each SNP, the test statistic of the Wald test for combining multiple test statistics from $K$ GWAS asymptotically follows the chi-squared distribution with $K$ df, and can be estimated by considering only the background correlation,

$$t_w = \mathbf{t}^\mathbf{T}\mathbf{\Psi}^{-1}\mathbf{t} \sim \chi^2_K \qquad (9)$$

The fact that the Wald test does not rely on $\Pi$ compensate the power loss of EbICoW when all elements of $|\Pi\text{-}\Psi|$ is small.

## Combining multiple GWASs

C-GWAS uses both EbICoW and Wald to improve the power of detecting SNPs with multi-trait effects. Here the key concepts involve the decision-making, subset identification, and distribution calibration. Below we present an algorithm with two consecutive modules to achieve these purposes. The first module is iterative EbICoW (i-EbICoW), which iteratively apply a decision-making algorithm to select subsets suitable for EbICoW, resulting in a set of EbICoW combined GWAS and potentially some individual GWAS, in such a way that all GWAS pairs from i-EbICoW are no longer suitable for applying EbICoW. The second module is a truncated Wald test (TWT) in combination with distribution calibration. TWT further analyze i-EbICoW results in a SNP-specific manner, which produce all possible SNP-specific subsets using a series of predefined thresholds. Then a series of multiple testing corrections, using *getNtest* and *Tippett*, are carried out for all intermediate results to ensure all threshold-based subsets have the same level of $\alpha$. The choice of best subset is made according to the levels of significance from TWT of all subsets after adjusting. Finally, *getCoef* guarantees that the final results from C-GWAS are fully adjusted for the number of GWASs and for all intermediate iterative steps, thus directly applicable using the traditional genome-wide significant threshold as our study-wide significance threshold.

## The first module: i-EbICoW

Theoretically, $|\pi\text{-}\psi|\in[0, 2]$. The more $|\pi\text{-}\psi|$ is deviated from 0, the higher the power of EbICoW than that of Wald and vice versa. We introduce a decision-making algorithm for the choice of using EbICoW by taking $|\pi\text{-}\psi|$ as one of the key parameters into consideration. In addition, when combining a large number of GWAS, the relationships between $\Pi$ and $\Psi$ tend to be complex. SNPs unlikely share the same $\Pi$ across all GWAS and $|\pi\text{-}\psi|$ may be sufficiently large only in a subset of GWAS. This requires finding the subsets best suitable for EbICoW, which is achieved using a data-driven approach by iteratively applying our decision-making algorithm.

Consider $K$ GWAS $\mathbf{G}\in(\mathbf{G_1}, \mathbf{G_2},\ldots,\mathbf{G_K})$, where $\mathbf{G_i}$ contains all summary statistics of the GWAS $k$, including the test statistic vector $\mathbf{T_k}$, the effect vector $\mathbf{B_k}$, and the $p$-value vector $\mathbf{P_k}$ of genome-wide SNPs, derived $\Psi$, $\Pi$, and $\mathbf{h}$ using functions $\Psi = \text{getPsi}(\mathbf{G})$, $\Pi = \text{getPi}(\mathbf{G})$, and $\mathbf{h} = \text{getH}(\mathbf{G})$. The *getPsi*, *getPi* and *getH* functions are described in details in the Estimation of $\psi$ and Estimation of $\pi$ and $\mathbf{h}$ sections. Define $F$ as minimum difference of $\pi$ and $\psi$ allowed for combined analysis in each iteration. Default of $F$ is set to 0.05. Set $U$=0, $U$ is a count of iterations in which *evaluate* returns TRUE, and $\mathbf{D}=|\Pi\text{-}\Psi|$, $\mathbf{D}$ is a $K$ by $K$ symmetric matrix of diagonal of 0 to reflect difference of $\pi$ and $\psi$ for each pair of GWAS. The process of i-EbICoW is described below:

while any element in $\mathbf{D} > F$ {

$(i, j) = \text{which}(\text{Max}(\mathbf{D}))$; # get the index of the maximal element of $\mathbf{D}$

$\mathbf{G_E} = \text{optimize}(\text{EbICoW}(\mathbf{G_i}, \mathbf{G_j}, \Psi, \Pi, \mathbf{h}))$;

$\mathbf{P_W} = \text{Wald}(\mathbf{T_i}, \mathbf{T_j})$; # using (9) for $\mathbf{t}$ of each SNP

$\mathbf{P_A} = \text{Min}(\mathbf{P_i}, \mathbf{P_j})$; # get the minimal $p$-value of each SNP

if evaluate$(\mathbf{G_E}, \mathbf{P_W}, \mathbf{G_A})$ is TRUE {

Remove$(\mathbf{G_i}, \mathbf{G_j})$ from $\mathbf{G}$; $\mathbf{G} = (\mathbf{G}, \mathbf{G_E})$;

$\Psi = \text{getPsi}(\mathbf{G})$; $\Pi = \text{getPi}(\mathbf{G})$; $\mathbf{D} = |\Pi\text{-}\Psi|$; $U = U+1$;

} else $\{D_{ij}=0\}\}$;

The *optimize* function (Supplementary Method B) is to select the optimal weight for EbICoW (5) to combine two GWAS where the effect correlation $\pi$ and effect vector $\mathbf{h}$ is chosen from three our proposed ones, $\mathbf{G_{all}} = \text{EbICoW}(\mathbf{G_i}, \mathbf{G_j}, \Psi,\Pi_{all},\mathbf{h_{all}})$; $\mathbf{G_{sig}} = \text{EbICoW}(\mathbf{G_i}, \mathbf{G_j}, \Psi, \Pi_{sig},\mathbf{h_{sig}})$; $\mathbf{G_{stb}} = \text{EbICoW}(\mathbf{G_i}, \mathbf{G_j}, \Psi,\Pi_{stb},\mathbf{h_{stb}})$. The proposal of the three $\pi$ and $\mathbf{h}$ is described in detail in the Estimation of $\pi$ and $\mathbf{h}$ section. The *evaluate* function make decision whether accept combined result from *optimize* via evaluating the power and robustness of $\mathbf{G_E}$ (Supplementary Method C). i-EbICoW returns $K\text{-}U$ newly obtained GWAS $(\mathbf{G_1}, \mathbf{G_2},\ldots,\mathbf{G_{K-U}})$. All pairs of updated $\mathbf{G}$ are no longer suitable for applying EbICoW and these results are further passed to the second step analysis.

## The second module: TWT and distribution calibration

TWT analysis is carried out on the $K\text{-}U$ newly obtained GWAS $(\mathbf{G_1}, \mathbf{G_2},\ldots,\mathbf{G_{K-U}})$ from i-EbICoW. Set a vector of thresholds as $\mathbf{r} = (r_1, r_2,\ldots, r_Q)$, where $r\in[1, 0)$ in the descending order. In default $r_q = 10^{-q/3}$ and $Q = 18$, where $\mathbf{q} = (1, 2,\ldots, Q)$. For each SNP $m\in\{1, 2,\ldots, M\}$, the vector of $p$-value and $T$ statistics are $\mathbf{p} = (P_{m,1}, P_{m,2},\ldots, P_{m,K-U})^\mathbf{T}$ and $\mathbf{t} = (T_{m,1}, T_{m,2},\ldots, T_{m,K-U})^\mathbf{T}$. The vector of combined $p$-values $\mathbf{cp} = (cp_1, cp_2,\ldots, cp_Q)^\mathbf{T}$ from a series of $\mathbf{r}$-based subsets is derived as $\mathbf{cp} = \text{TWT}(\mathbf{p},\mathbf{t})$, where TWT is a function of $\mathbf{p}$ and $\mathbf{t}$ as defined below:

For each element $r_q$ in $\mathbf{r}$ {

$\mathbf{id} = \text{which}(\mathbf{p}r_q)$;

if (length($\mathbf{id}$)=0) $\{cp_q = \text{NA}\}$;

else $\{cp_q = F_{\chi^2_{\text{length}(\mathbf{id})}}(\mathbf{t_{id}^T}(\mathbf{\Psi_{id,id}})^{-1}\mathbf{t_{id}})\}$;

Return $\mathbf{cp}$;

$F_{\chi^2_Y}(x)$ denotes the cumulative distribution function of a chi-squared random variable $x$ with $Y$ degrees of freedom. The minimal $p$-value of $\mathbf{p}$ can be regarded as a one-element subset. Therefore, $cp_{Q+1} = \text{Min}(\mathbf{p})$. After applying TWT to all SNPs on the genome, we have a $p$-value matrix $\mathbf{CP}$ with $M$ rows (the number of SNPs) and $Q+1$ columns (the number of preset thresholds add one of minimum), where $\mathbf{CP} = [\mathbf{CP_1}, \mathbf{CP_2},\ldots,\mathbf{CP_M}]^\mathbf{T}$.

The next idea is for each row $\mathbf{CP_m}$ to find the best subset under the threshold $r_q$, which corresponds to the minimal $p$-value in this row. However, the results from this selection procedure deviate from any known distribution. Multiple testing correction and distribution calibration is necessary to guarantee all columns of $\mathbf{CP}$ and final best-subset-based results are comparable at the same $\alpha$. To achieve this we firstly adjusted all columns empirically in $\mathbf{CP}$ using $NS$ rounds of simulations as below:

For $ns$ in 1 to $NS$ $\{\mathbf{t_{simu}} \sim N(0,\Psi)$; $\mathbf{p_{simu}} = F_{\chi^2_1}(\mathbf{t_{simu}} \circ \mathbf{t_{simu}})$; $\mathbf{SP}_{ns} = \text{TWT}(\mathbf{p_{simu}}, \mathbf{t_{simu}})\}$;

For $q$ in 1 to $Q+1$ $\{\mathbf{CP_q} = \text{Tippett}(\mathbf{CP_q},\text{getNtest}(\mathbf{SP_q}))\}$;

$\mathbf{t_{simu}}$ is a random vector of length $K\text{-}U$ generated from multi-variable standard normal distribution with updated $\Psi$ (after i-EbICoW) in each round of simulations, and $\mathbf{SP}$ is $p$-value matrix with $NS$ rows and $Q+1$ columns containing all simulation results. Multiple testing correction methods *Tippett* and *getNtest* function used here is detailed in Supplementary Method D. With the adjustment described above, the best subset can be obtained from the adjusted $\mathbf{CP}$. Take the minimum of each $\mathbf{CP_m}$ as the final output $\mathbf{minCP} = (\text{min}(\mathbf{CP_1}), \text{min}(\mathbf{CP_2}),\ldots, \text{min}(\mathbf{CP_M}))$ with study-wide significance, we further adjust $\mathbf{minCP}$ to guarantee its null distribution and $\alpha$ are same as any conventional GWAS, i.e., uniform distribution and $\alpha = 5 \times 10^{-8}$ respectively. The selection of minimal $p$-value among the adjusted $p$-values in all combinations obtained by TWT inevitably results in a non-

uniform null and the inflation of this non-uniform null is quantile specific in nature. In order to make all final $p$-values from C-GWAS follow the uniform distribution under the null, we obtain the non-uniform null via the same $NS$ rounds of simulations and then calibrate the simulated non-uniform null to a uniform distribution using a quantile-based method (*getCoef* function, Supplementary Method D) as below:

For $q$ in 1 to $Q$+1 $\{\mathbf{SP}_q = \text{Tippett}(\mathbf{SP}_q, \text{getNtest}(\mathbf{SP}_q))\}$;

$$\mathbf{minCP} = \mathbf{minCP} \circ \text{predict}(\text{rank}(\mathbf{minCP})/M, \text{getCoef}(\mathbf{minSP}));$$

This method guarantees a uniform null for any arbitrarily defined statistics as long as the true null can be obtained via simulations. Therefore, the final **minCP** from C-GWAS is directly comparable to the $p$-values from a standard GWAS where the conventional $5 \times 10^{-8}$ significance threshold is applied. The *getCoef* is generic and we also use it for correcting the minimal $p$-values from MinGWAS. The empirical calibration is sufficiently accurate when $NS$ is large. We suggest use at least 100 times of independent SNP number for $NS$. In C-GWAS, the default of $NS$ is set to $10^8$.

## Estimation of Inflation, getI(G)

Accurate estimation of inflation is a prerequisite for accurate estimation of $\pi$ and $\mathbf{h}$. We propose a fast and accurate method for estimating inflation. Compared with the genomic control method, our method is less likely affected by the presence of true effects, thus less conservative. Compared with the LD score method described by Bulik-Sullivan et al.[22], our method has the same level of estimate error with higher computation efficiency because it does not require LD information, as detailed below.

Following the polygenicity model described in Bulik-Sullivan et al.[22], we modeled the test statistics $\mathbf{T}$ from a GWAS using a mixed distribution of three normal distributions, including the estimated error $N(0, e)$, inflation induced by genetic drift assumed to be normal distributed $N(0, I)$, and true effect assumed to be normal distributed $N(0, \Lambda)$. We additionally assume that $p$ is the proportion of the SNPs (including those in LD) having true effects. For a SNP $m$, where $m \in \{1, 2, ..., M\}$, we have

$$T_m \sim \begin{cases} N(0,e) + N(0,I) & \text{The probability is } 1 - p \\ N(0,e) + N(0,I) + N(0,\Lambda) & \text{The probability is } p \end{cases}$$

The moments of $\mathbf{T}$ can be expressed by $I$, $\Lambda$ and $p$,

$$E(T_m^2) = 1 + I + p\Lambda \qquad (10)$$

$$\text{Var}(T_m^2) = 2\left(E\left(T_m^2\right)\right)^2 + 3\Lambda^2 p(1 - p)$$

where $\mathbf{T}$ is known for individual GWASs, and $I$, $\Lambda$, $p$ are unknown. Our expression of $E(T_m^2)$ here corresponds to the expression of $Nh^2/M + Na + 1$ as described in Bulik-Sullivan et al.[22]. Note that the inflation term $Na$ is simplified as $I$ in (10) and true effect term $Nh^2/M$ is simplified as $p\Lambda$ in (10). Next, $I$ (or inflation factor $s = I+1$) can be determined using an iterative search algorithm with the moments of $\mathbf{T}$ and a series of quantiles of $\mathbf{T} \circ \mathbf{T}$ (Supplementary Method E). All GWAS can be pre-adjusted using GWAS-specific $s$ ($\mathbf{T_{adj}} = \mathbf{T}/\sqrt{s}$) before C-GWAS analysis.

## Estimation of $\psi$, getPsi(G)

Accurate estimation of $\psi$ is a prerequisite for combining correlated traits. We propose a fast and accurate method to estimate $\psi$. Compared with the methods which directly compute correlations using GWAS statistics, such as Zhu et al.[2] and Bolormaa et al.[1], our method is less likely affected by the true effect. Compared with MTAG, which

estimate covariance of the estimate error using LDSC[34], our method has a similar level of estimate error but does not require LD information, thus more computationally efficient, especially for the analysis of a large number of GWAS.

The idea of our method is similar to that of Park et al.[7], which select a set of marginally insignificant SNPs from two GWAS. Our approach selects a set of insignificant SNPs from the joint distribution of two GWAS, thus more tightly fitting the true null. This is achieved using an iterative optimization algorithm. Consider $T$ statistics of two GWAS for genome-wide SNPs $\mathbf{T_1}$ and $\mathbf{T_2}$, let $\psi_0$ be the correlation between $\mathbf{T_1}$ and $\mathbf{T_2}$ as $\psi_0 = \text{cor}(\mathbf{T_1}, \mathbf{T_2})$, $\psi_1 = 0$, and $\varepsilon = 10^{-4}$. The process of the *getPsi* is described as below,

while $|\psi_0 - \psi_1| > \varepsilon$ {
$\psi_0 = \psi_1$; $\boldsymbol{\Psi_0} = [(1, \psi_0), (\psi_0, 1)]$;
for $m$ in 1 to $M$ $\{CT_m = (T_{m,1}, T_{m,2})\boldsymbol{\Psi_0}^{-1}(T_{m,1}, T_{m,2})^T\}$;
$\mathbf{id} = \text{which}\left(F_{\chi_2^2}\mathbf{CT} > 0.5\right)$;
$\psi_1 = \text{cor}(\mathbf{T_{id,1}}, \mathbf{T_{id,2}})\}$;
$\psi$ is estimated as $\psi_1$. In a C-GWAS analysis, $\boldsymbol{\Psi}$ is derived only once and does not change. Therefore, $\boldsymbol{\Psi}$ is the same for all SNPs throughout the C-GWAS analysis.

## Estimation of $\pi$ getPi(G,$\Psi$) and h getH(G,$\Psi$,$\Pi$)

Let $\mathbf{B}$ and $\mathbf{Z}$ both be a $K$ by $K$ covariance matrix of $\boldsymbol{\beta}$ and $\mathbf{t}$ among the genome-wide SNPs, where $B_{ij} = E(\beta_i\beta_j)$ and $Z_{ij} = E(t_it_j)$. $\mathbf{H}$ and $\mathbf{C}$ denote the covariance matrix of $\boldsymbol{\delta}$ and $\boldsymbol{\beta_{null}}$ ($\boldsymbol{\beta}$ under the null) among the genome-wide SNPs as described above. The basic idea of estimating $\pi$ and $\mathbf{h}$ is similar to the estimation of covariance of true effect in MTAG, i.e., to estimate the expected overall difference ($\boldsymbol{\Pi_{all}}$ and $\mathbf{h_{all}}$) between observed effects and the estimation error, i.e., $\mathbf{H} = \mathbf{B} - \mathbf{C}$. For each SNP, we assume $\boldsymbol{\Psi}$ and $\boldsymbol{\Pi}$ is fixed but $\mathbf{h}$ can vary with different $\boldsymbol{\sigma}$ to keep a unified null distribution while accounting for potential heterogeneity of the statistical power of SNPs in different GWASs. Therefore,

$$\mathbf{H} = \mathbf{B} - \mathbf{C} = (\mathbf{Z} - \boldsymbol{\Psi}) \circ \boldsymbol{\sigma}\boldsymbol{\sigma}^{\mathbf{T}}; \quad \text{let } \Pi_{ij}^* = H_{ij}/\sqrt{H_{ii}H_{jj}} \text{ and } \boldsymbol{\Pi_{all}} = \boldsymbol{\Pi}^*$$

$$\mathbf{h_{all}} = \text{sign}\left(\boldsymbol{\Pi_{all}} - \boldsymbol{\Psi}\right)_k \mathbf{H} = \text{sign}\left(\boldsymbol{\Pi_{all}} - \boldsymbol{\Psi}\right)_k((\mathbf{Z} - \boldsymbol{\Psi}) \circ \boldsymbol{\sigma}\boldsymbol{\sigma}^{\mathbf{T}})$$

These estimators fits best under polygenicity, i.e., all SNPs share the same $\boldsymbol{\Pi}$. In deviation of polygenicity, e.g., major gene effect, or when dealing with low-powered GWAS, the power may not be optimal. To overcome this limitation, we provide an alternative estimator ($\boldsymbol{\Pi_{sig}}$ and $\mathbf{h_{sig}}$) which focus on the SNPs with more significant effects rather than genome-wide SNPs,

$$\mathbf{H} = \mathbf{B} - \mathbf{C} = \left(\mathbf{Z} - \boldsymbol{\Psi_{nosig}}\right) \circ \boldsymbol{\sigma}\boldsymbol{\sigma}^{\mathbf{T}}; \quad \text{let } \Pi_{ij}^* = H_{ij}/\sqrt{H_{ii}H_{jj}} \text{ and } \boldsymbol{\Pi_{sig}} = \boldsymbol{\Pi}^*$$

$$\mathbf{h_{sig}} = \text{sign}\left(\boldsymbol{\Pi_{sig}} - \boldsymbol{\Psi}\right)_k \mathbf{H} = \text{sign}\left(\boldsymbol{\Pi_{sig}} - \boldsymbol{\Psi}\right)_k((\mathbf{Z} - \boldsymbol{\Psi_{nosig}}) \circ \boldsymbol{\sigma}\boldsymbol{\sigma}^{\mathbf{T}})$$

Where $\boldsymbol{\Psi_{nosig}}$ is the covariance matrix of the non-significant SNPs among $K$ GWASs. Here the 'significant' is defined as $p$-value of Wald test using (9) below a predefined threshold $L$. These estimators is expected to perform better in the presence of major gene effect and when dealing with low-powered GWAS.

Alternatively, to improve the robustness of estimation, when the estimation error of $\mathbf{H}$ is large, setting $\boldsymbol{\Pi}$ to a square matrix of 1 ($\boldsymbol{\Pi_{stb}}$) or assuming the same $\boldsymbol{\delta}$ as usually done in the most of previous multi-trait analysis method may provide stable or higher power. In this case, $\mathbf{h_{sig}}$ can be derived as $\sum_{k=1}^{K}(\sigma_k\sqrt{Z_{kk} - 1})\mathbf{h}^*$, where $\mathbf{h}^* = (\sigma_1\sqrt{Z_{11} - 1}, \sigma_2\sqrt{Z_{22} - 1}, ..., \sigma_K\sqrt{Z_{KK} - 1})$. Because the choice of $\boldsymbol{\Pi}$ and $\mathbf{h}$ does not influence $\alpha$ but has an effect on power, the selection of $\boldsymbol{\Pi}$ and $\mathbf{h}$ is data-specific and can be achieved empirically using a data-driven approach (details see combining multiple GWASs section). Similar to $\boldsymbol{\Psi}$, $\boldsymbol{\Pi}$ is the same for all SNPs. However, different from $\boldsymbol{\Psi}$, $\boldsymbol{\Pi}$ is

continuously updated in each iteration of i-EbICoW so that different GWAS subsets may have different $\mathbf{\Pi}$. For example, consider 5 GWASs with estimated $\mathbf{\Psi}$ and $\mathbf{\Pi}$, the $\mathbf{\Psi}$ of the GWAS subset 1,2,3 used in the full process of C-GWAS is $\mathbf{\Psi}_{[1\text{-}3, \, 1\text{-}3]}$ while the $\mathbf{\Pi}$ of this subset after updated in i-EbICoW may be different from initial $\mathbf{\Pi}_{[1\text{-}3, \, 1\text{-}3]}$.

## Simulations

We conducted extensive simulations to investigate the performance of C-GWAS under a variety of scenarios. For simulation scenarios 1–9, the core of our simulations is to simulate the test statistics of $N$ independent SNPs on $K$ GWAS (**simT**, a matrix with $N$ rows and $K$ columns) under the null by generating $K$ null vectors of length $N$, which follow the multivariate standard normal distribution with a predefined background correlation matrix $\mathbf{\Psi}$, so that for one simulated vector of test statistics **simt**, we have $\mathbf{simt} \sim N(0, \mathbf{\Psi})$. For a GWAS $\mathbf{simT}_k$, a vector of test statistics which follow normal distribution with variance $I$ is added to simulate inflation, so for a SNP $simT_{nk}$ in GWAS $\mathbf{simT}_k$, we have $simT_{nk} \sim N(0,1) + N(0,I)$. To add true effects, a proportion ($p$) of the $N$ SNPs were set to have effects by adding a matrix of test statistics which follow multivariate normal distribution with covariance matrix $\mathbf{\Pi} \circ \theta\theta^T$, where $\theta$ is a preset vector of standard deviation of each column of the added effect matrix, so that we have

$$simT_{n1} \sim N(0, \mathbf{\Pi} \circ \theta\theta^T) + N(0, \mathbf{\Psi}), \quad \text{where } n1 \in \{1, 2, \ldots \lfloor pN \rfloor\}$$

$$simT_{n2} \sim N(0, \mathbf{\Psi}), \quad \text{where } n2 \in \{\lceil pN \rceil, \lceil pN \rceil + 1, \ldots, N\}$$

The mean $\chi^2$ statistics ($\mathrm{E}(\chi^2)$) of simulated GWAS $k$ can be derived as $1 + p\theta_k^2$. The ratio of variance of true effect $\delta$ can be set via multiplying preset ratio between elements of $\theta \circ \theta$ and vectors of squared standard errors $\boldsymbol{se} \circ \boldsymbol{se}$, i.e., $\frac{\mathrm{Var}(\delta_i)}{\mathrm{Var}(\delta_j)} = \frac{\theta_i^2 se_i^2}{\theta_j^2 se_j^2}$. The parameter grids in different nine scenarios are specified in Supplementary Method F, and the core was replicated 1,000 times for every cell of the grid in 1–9 scenarios.

For scenario 10 assess the performance of C-GWAS and MTAG in combining GWASs of overlapping samples, we used the real chromosome 1 genotype data (526,822 SNPs) of 10,000 participants randomly selected from the Rotterdam Study (total 14,926 participants). We simulated a normally distributed variable as the phenotype in such a way that 10% of the phenotypic variance is explained by 1% of all SNPs, whose true effects follow a normal distribution. We created three sub-datasets according to different configurations of sample overlapping, (1) non-overlapping with sizes of 2000, 3000, and 5000; (2) partially overlapping with sizes of 5000, 7000, and 8000; and (3) almost complete overlapping of 9950 each. We compared the results of C-GWAS and MTAG in combining the three GWASs in sub-datasets with the GWAS in all samples.

## C-GWAS specifications in real applications

The *getI* and *getPsi* functions use unlinked SNPs via multiple rounds of systematic sampling. For input GWASs with $\mathrm{E}(\chi^2) < 1.001$, C-GWAS forces $\mathrm{E}(\chi^2) = 1.001$ to avoid failure of *getH*. To avoid abnormal values when solving large $\mathbf{\Psi}$ in the presence of strong collinearity, GWAS pairs with $\psi^2 > 0.5$ (default) are forced combined using *optimize* function regardless to *evaluate* function.

## C-GWAS implementation and performance

C-GWAS was implemented as a user-friendly, publicly accessible, operating system independent and parallel R package called "C-GWAS", https://github.com/Fun-Gene/CGWAS. The idea of improving computational efficiency is to locate all independent parts from the whole serial calculations and parallelize them using R package

"foreach" and "doParallel". In general, all independent loops are parallelized. For the computational hotspot *getPsi*, since it is nested in the iterative decision-making structure of i-EbICoW, we designed fine-grained parallelization within each iteration for calculating $\psi$ between multiple GWASs. For the hotspot *getNtest* of the TWT module, because $NS$ simulations are independent, the parallelization is designed at a coarse-grained level. For controlling peak memory during parallelization, 1) we minimized the duplicated data between the sub-threads and main thread, optimized the balance between the overall computational efficiency and the loads of sub-threads based on a hierarchical load distribution; 2) reduced the number of slave threads for simple calculations but requesting large memory.

We assessed the performance of C-GWAS and MTAG on a machine with 72 cores Intel Xeon CPU at 2.30GHz and 256 GB RAM, using simulated data (6 million SNPs; 5, 10, 20, 30, 40, and 80 GWASs; 1, 2, 4, 8, and 16 paralleled threads). Scenarios in each configuration were replicated testing for 3 times, and the mean performance was reported in Supplementary Table 2.

## C-GWAS analyses in 78 facial shape GWASs

We applied C-GWAS to combine the summary statistics of 78 GWAS of facial traits, which were conducted in 10,115 individuals of European decent as described in Xiong at al[4]. Prior to C-GWAS, we dropped SNPs with sample sizes smaller than 60% of the total sample sizes from 7,029,494 to 5,308,962. This is because a reduced sample size in a subset of SNPs may increase the estimation error of $\mathbf{\Pi}$ and thus hamper the power of EbICoW. A simulation analysis imbedded in C-GWAS was conducted to obtain the true null of our C-GWAS application. In brief, the null summary statistics of 78 GWAS were simulated, each consisting of 1,000,000 unlinked SNPs without true effect, where the background correlations were simulated according to the $\mathbf{\Psi}$ estimated from the real GWAS data using the *getPsi* function. The same GWAS sets and corresponded $\mathbf{w}^T$ of each combination in i-EbICoW of real GWAS data were used to combine simulation data. This step derived the same EbICoW GWAS with the real application but using simulated null. Next, C-GWAS obtain the true null by applying TWT and distribution calibration as described above and finally adjust the application $p$-values using the outcome of *getCoef* from simulation to ensure that C-GWAS results are directly comparable to a standard GWAS. To calculate genome-wide significant threshold for each i-EbICoW combination in C-GWAS final $p$-values, we derive multiple testing and calibration burden estimated in *getNtest* and *getCoef* function as 27.13 and 2.22 respectively (Supplementary Method D), which results the threshold as $5 \times 10^{-8} \times 27.13 \times 2.22 = 3 \times 10^{-6}$. Under this threshold, C-GWAS and MinGWAS results are provided in Supplementary Data 2. Regional lead SNPs (56) were selected based on the FUMA clumping algorithm[35] with default parameter as detailed in Supplementary Data 3. Univariate LD score regression[22] was applied to z-transformed GWAS $p$-values, $\mathbf{z} = \sqrt{F_{\chi_1^2}^{-1}(1 - \mathbf{p})}$. The LD score intercepts of C-GWAS and MinGWAS were estimated as 0.985 (0.007) and 0.998 (0.006) respectively. The gain of power for C-GWAS was estimated according to the increase in the mean $\chi^2$ statistic method described in Turley et al.[3]. In brief, the power ratio between two GWASs can be derived as $\frac{\mathrm{E}(\chi^2_{GWAS1}) - \mathrm{E}(\chi^2_{null})_{GWAS1}}{\mathrm{E}(\chi^2_{GWAS2}) - \mathrm{E}(\chi^2_{null})_{GWAS2}}$. In our study, $\mathrm{E}(\chi^2_{C-GWAS}) = 1.142$ and $\mathrm{E}(\chi^2_{MinGWAS}) = 1.1$, so that $\frac{1.142 - 0.985}{1.1 - 0.998} = 1.54$.

## Replication of C-GWAS findings and previous GWAS findings

The RS is a population-based cohort study of 14,926 participants aged 45 years and older, living in the same suburb of Rotterdam, the Netherlands[36]. The present study includes 1,174 participants of Dutch European ancestry, for whom high-resolution 3dMDface digital photographs were taken. Note that these samples have not been used in the previous GWAS[4]. Genotyping was carried out using the Infinium II

HumanHap 550K Genotyping BeadChip version 3 (Illumina, San Diego, California USA). Microarray-based genotyping according to the manufacturer's instructions was performed at Erasmus MC[37]. All SNPs were imputed using MACH software (www.sph.umich.edu/csg/abecasis/MaCH/) based on the 1000-Genomes Project reference population information[38]. After all quality controls, the current study included a total of 6,886,439 autosomal SNPs (MAF > 0.01, imputation R2 > 0.8, SNP call rate > 0.97, HWE > 0.0001).

The Rotterdam Study has been approved by the Medical Ethics Committee of the Erasmus MC (registration number MEC 02.1015) and by the Dutch Ministry of Health, Welfare and Sport (Population Screening Act WBO, license number 1071272–159521 PG). The Rotterdam Study has been entered into the Netherlands National Trial Register (NTR; www.trialregister.nl) and into the WHO International Clinical Trials Registry Platform (ICTRP; www.who.int/ictrp/network/primary/en/) and under shared catalog number NTR6831. All participants provided written informed consent to participate in the study and to have their information obtained from treating physicians.

The raw 3D facial images of participates were acquired using a 3D photographic scanning system manufactured by 3dMD (http://www.3dmd.com/). Participants were asked to keep their mouths closed and adopt a neutral expression during the acquisition of the 3D scans. Software package MeshMonk[39] was then used to derive the 13 facial landmarks. The description of these 13 landmarks are in Supplementary Table 3. Association tests were conducted between the 78 3D facial distance measurements (after scaled GPA) and 6,886,439 SNPs. GWAS was performed using linear regression under the additive genetic model while adjusting for sex, age, and the first four genomic PCs. C-GWAS analyses was then conduct use default parameters.

In addition, we reached out and obtained the GWAS summary statistics[5] from 2,774 individuals of European descent from the Pennsylvania State University (PSU) cohort ($N$ = 1990) and the Indiana University-Purdue University Indianapolis (IUPUI) cohort ($N$ = 784), which have not been used in our previous study[4] and are thus not included in our C-GWAS discovery dataset used here. The data contains 63 sets of GWAS summary statistics corresponding to the 63 facial segments. In addition, we downloaded the summary statistics from a recent facial variation GWAS of Chinese population[6], which included 9674 Chinese individuals from 3 cohorts, i.e., the National Survey of Physical Traits (NSPT), the Northern Han Chinese (NHC) and the Taizhou Longitudinal Study (TZL). The data also contains 63 sets of GWAS summary statistics for 63 facial segments. We then carried out our replication analysis for the 56 SNPs in RS, PSU+IUPUI, and NSPT+NHC+TZL.

We select 56 lead SNPs C-GWAS $p$-value in RS C-GWAS result as replicated evidence from RS. We use Bonferroni and Fisher's combined test to combine the evidence of replication and account for the trait differences between our landmarks-based study and the previous facial segment-based studies. In brief, we first adjust the minimal $p$-value of the 63 segment-based $p$-values using Bonferroni correction of the numbers of independent traits as reported in corresponding studies. We then use Fisher's combined test to combine the RS $p$-value derived from C-GWAS, the adjusted minimal $p$-value from PSU+IUPUI, and the adjusted minimal $p$-value from NSPT+NHC+TZL. In short, the test statistic of the Fisher's combined test $T_F$ of $N$ independent tests with $p$-value $p_n$ follows a chi-squared distribution of $2N$ degrees of freedom as $T_F = -2\sum_{n=1}^{N}\log(p_n) \sim \chi_{2N}^2$. Replication results are detailed in Supplementary Data 3. For 56 independent observations, the $p$-value of uniform null distribution at their expected quantile were generated from beta distribution. For 327 SNPs that have been associated with facial variation in previous GWASs and also available in our GWAS data, we looked up their $p$-values from our C-GWAS and MinGWAS analysis of 10,115 individuals.

## Comparison with other multi-trait analysis methods using facial data

C-GWAS and MTAG have an important similarity in distinguishing the effect correlation $\Pi$ (true effect $\Omega$ in MTAG) from the background correlation $\Psi$ (estimate error $\Sigma$ in MTAG) with the advantage in better estimating true null, whereas other methods do not. In addition, when $\Pi$ and $\Psi$ are different, both methods provide additional statistical power, while the other methods do not. It is therefore desirable to compare the statistical performance of the two methods. C-GWAS and MTAG analyses were carried out using GWAS summary statistics generated previously, which consisted of 10,115 individuals of European decent. Because MTAG cannot complete the analysis of 78 facial GWASs within our computational capacity, we focused on a subset of 30 GWAS covering the left side of the face. Because MTAG generated 30 GWAS outcomes (one outcome per trait), we selected the minimal $p$-values and adjusted them using the *getCoef* function. C-GWAS setting was the same with those used in all 78 GWAS application. To calculate genome-wide significant threshold for each i-EbICoW combination in C-GWAS final $p$-values, we derive multiple testing and calibration burden estimated in *getNtest* and *getCoef* function as 12.44 and 2.17 respectively (Supplementary Method D), which results the threshold as $5 \times 10^{-8} \times 12.44 \times 2.17 = 1.35 \times 10^{-6}$. Under this threshold, C-GWAS and MTAG results are provided in Supplementary Data 4. We additionally compared C-GWAS with four competing statistics, including mixAda, mixFisher and mixTippett proposed by Liu et al. using R package 'MPAT'[23] and $S_{Het}$ proposed by Zhu et al. using R script 'CPASSOC'[2]. We also looked up the 327 previously established SNPs in C-GWAS and results from all competing methods described above. Note that due to SNP filtering of MTAG, only 271 SNPs retained for comparison.

## Polygenetic score and facial variance explained

Multivariable and polygenic score analyses were conducted in the same 1,174 participates from RS which described above. 57 lead SNPs in C-GWAS, 57 and 17 lead SNPs in MinGWAS were involved in analyses below. We calculated PRS using the selected $n$ SNPs and the effect sizes estimated from the GWAS summary of 10,115 individuals, that is $\sum_{i=1}^{n}\beta_i A_i$. Then proportion of phenotype explained for 78 facial traits were assessed by $r^2$ between MinGWAS based and C-GWAS based PRS and sex-, age- adjusted face phenotypes. 95% confidence intervals of $r^2$ were obtained from 1000 times randomly sampling with replacement. PRS z score test were then conducted by cooperating $r^2$ value and standard deviation of sampling results. Next, we calculated PRS for AFR, EUR, EAS individuals of the 1000 Genomes Project using 57 lead SNPs in C-GWAS, 17 lead SNPs in MinGWAS and effect sizes as used above.

## Estimation of multi-trait effects

We developed a statistic to quantify multi-trait effect (MTE) for the positive findings from C-GWAS. The idea is to use the skewness of the independent phenotypic effects of each SNP to represent its multi-trait effect, i.e., the more independent phenotypes a SNP is associated with (less skew), the larger its multi-trait effect is. The skewness statistic of a vector $\mathbf{X}$ can be estimated using the skew function, $\mathrm{Skew}(\mathbf{X}) = E(\mathbf{X} - E(\mathbf{X}))^3 / (E(\mathbf{X} - E(\mathbf{X}))^2)^{3/2}$, as proposed previously[40]. However, because the phenotypes are correlated due to background correlations, steps to remove background correlations are necessary. To achieve this, we firstly construct an uncorrelated null space of eigen vectors set $\mathbf{V}$ on $\Psi$, so that all vectors in this space have no background correlations under the null. In our application, we derived top 26 eigen vectors with decreased eigen value from 78 eigen vectors of $\Psi$ to form $\mathbf{V}$ for eliminating the noise from the eigen vectors with small eigen value. Number of derived eigen vectors was ascertained by keeping the accumulated eigen value over 95% of total 78. Next, for each significant

SNP, its phenotypic effects $\mathbf{t} = (t_1, t_2, \ldots, t_{78})^T$ is projected to the null space as $\mathbf{pt} = (pt_1, pt_2, \ldots, pt_{26})^T$ and then is scaled using the eigen value $\mathbf{g}_{(1-26)}$ to ensure that they are comparable (same variance) under the null, i.e., $pt_i = \mathbf{t}^T \mathbf{V}_i / \sqrt{g_i}$, where $i \in \{1, 2, \ldots, 26\}$. Finally, we calculate the skewness of squared projected effects (eliminate the impact of sign, focus on absolute effect) for each significant SNP, then centralize these skewness values to that under the null. Noted that the squared projected effects under the null follow the chi-squared distribution with one degree of freedom. Therefore, the skewness value under the null can be directly calculated as $E(\chi_1^2 - E(\chi_1^2))^3 / (E(\chi_1^2 - E(\chi_1^2))^2)^{3/2} = (E(\chi_1^2)^3 - 3E(\chi_1^2)\mathrm{Var}(\chi_1^2) - (E(\chi_1^2))^3) / \mathrm{Var}(\chi_1^2)^{3/2} = (15 - 6 - 1)/2\sqrt{2} = 2\sqrt{2}$. The inverse of the centralized skewness of each SNP is considered as a quantification of its multi-trait effect, MTE=$2\sqrt{2}$-Skew($\mathbf{pt} \circ \mathbf{pt}$). MTE < 0 indicates insufficient evidence supporting the presence of multi-trait effect. MTE values for each C-GWAS lead SNP are in Supplementary Fig. 15.

### Gene ontology analysis

To reveal and compare the potential functional roles of the C-GWAS and MinGWAS findings, we investigate the enrichment of nearby genes from MinGWAS and C-GWAS loci in three otology databases, including Gene Ontology (GO), Human Phenotype Ontology (HPO) and Mouse Phenotype Ontology (MPO). The gene sets used for C-GWAS (total 83 genes from 56 loci) and MinGWAS (total 24 genes from 17 loci) represented the unions of three non-exclusive sets, including the closest genes to regional lead SNPs (57 for C-GWA, 17 for MinGWAS), annotated genes from 'SNP2GENE' function in FUMA[35] (64 for C-GWAS, 12 for MinGWAS), and cis-regulated genes proposed by GREAT[41] based on lead SNPs (70 for C-GWAS, 22 for MinGWAS). The enrichment analysis was performed using the R package 'clusterProfiler'[42] in GO and using GREAT in HPO and MPO. All Bonferroni adjusted significant ($p<0.05$) enriched terms are in Supplementary Data 5. The resultant GO terms were classified into 4 groups according to functionality, 'morphogenesis', 'development', 'differentiation', and 'regulation', whereas all the resultant HP and MP terms could be classified into one group 'abnormality'.

### Colocalization analysis

We performed Bayesian colocalization analysis between C-GWAS loci and eQTL dataset from GTEx[43] including 22 tissues from brain, skin, adipose, muscle skeletal and glands using R package 'coloc'[44]. This method evaluates whether a shared causal variant is responsible in both GWAS and gene expression in a genomic region of interest. Each C-GWAS locus was defined as a region with upstream and downstream 500kb from the regional lead SNP. The prior probabilities are set by default. A posterior probability was obtained for every gene around a region and a high posterior probability (PP4>0.7) suggested strong evidence of colocalization. Colocalizations in all 22 tissues of the loci with high posteriori probability (PP4>0.7) in at least one tissues are in Supplementary Data 6. Colocalization plots were generated using R package 'locuscomparer'[45].

### Analysis of CNCC regulation network

The regulatory network of CNCC was downloaded at https://github.com/AMSSwanglab/hReg-CNCC. We used ISTAT[28] to compute the enrichment of detected loci in the regulatory element of CNCC. The loci associated regulatory was obtained according to their intersection with regulatory elements proposed in regulatory network of CNCC[27]. For every locus, we checked every TF-RE-TG triplet and extracted it if the RE was intersected with this locus. Extracted triplet based on C-GWAS and MinGWAS loci are in Supplementary Data 7. All loci associated TF-RE-TG triplets were pooled to form the loci associated regulatory subnetwork of CNCC.

### Reporting summary

Further information on research design is available in the Nature Portfolio Reporting Summary linked to this article.

## Data availability

C-GWAS *p*-values of the study-wide suggestively significant SNPs are provided in Supplementary Data 2. Full C-GWAS summary statistics are publicly available via figshare at https://doi.org/10.6084/m9.figshare.21559086. Full GWAS summary statistics of the 78 facial traits used in discovery phase of C-GWAS application are available on GWAS Catalog with the access number 31763980. The summary statistics of the Pennsylvania State University (PSU) and the Indiana University-Purdue University Indianapolis (IUPUI) are available on GWAS Catalog with the access number 33288918. The summary statistics of the National Survey of Physical Traits (NSPT), the Northern Han Chinese (NHC) and the Taizhou Longitudinal Study (TZL) are available on the National Omics Data Encyclopedia with NODE number OEP002283. The cis-eQTL results in 22 tissues were downloaded from GTEx V7 database at https://www.gtexportal.org/home/datasets. The regulatory network of CNCC is publicly available at https://github.com/AMSSwanglab/hReg-CNCC.

## Code availability

CGWAS is implemented as an open-source R package available at https://github.com/Fun-Gene/CGWAS.

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

## Acknowledgements

The authors are grateful for the dedication, commitment and contribution of the study participants, the general practitioners, pharmacists, and the staff from the Rotterdam Study. The Rotterdam Study is supported by the Erasmus MC; the Erasmus University Rotterdam; the Netherlands Organization for Scientific Research (NWO); the Netherlands Organization for Health Research and Development (ZonMw); the Research Institute for Diseases in the Elderly (RIDE); the Netherlands Genomics Initiative (NGI); the Ministry of Education, Culture and Science; the Ministry of Health Welfare and Sport; the European Commission (DG XII); and the Municipality of Rotterdam. The generation and management of GWAS genotype data for the Rotterdam Study were executed by the Human Genotyping Facility of the Genetic Laboratory of the Department of Internal Medicine, Erasmus MC. We thank Susan Walsh and Mark Shriver for sharing with us for replication purposes in this study their published data[5] from the Indiana University-Purdue University Indianapolis (IUPUI) and Pennsylvania State University (PSU) cohorts. The authors of Zhang et al.[6] for making their GWAS summary statistics publicly available allowing us to use them here for replication purposes. FL was supported by the Strategic Priority Research Program of Chinese Academy of Sciences (Grant Nos. XDB38010400, XDC01000000), Shanghai Municipal Science and Technology Major Project (Grant No. 2017SHZDZX01), National Natural Science Foundation of China (NSFC) (81930056), Science and Technology Service Network Initiative of Chinese Academy of Sciences (KFJ-STS-QYZD-2021-08-001, KFJ-STS-ZDTP-079).

## Author contributions

Z.X., M.K., and F.L. designed and drafted the work. Z.X., X.G., and Y.C. developed mathematical formulations, performed data analyses, developed software package, visualized, and interpreted results. Z.F., S.P., and H.L. contributed to data analysis. A.G.U., T.N., A.I., F.R., M.G., Y.W., M.K., and F.L. provided the data resource. Y.W. critically revised all mathematical formulas. All authors approved the submission of the final version of the article for publication.

## Competing interests

The authors declare no competing interests
