## [Peer Review File · Nature Communications]

Combining Genome-wide Association Studies Highlights Novel Loci Involved in Human Facial VariationREVIEWER COMMENTS

Reviewer #1 (Remarks to the Author):

The authors proposed C-GWAS, a new method for combining GWAS summary statistics of multiple traits. They performed extensive simulations to show C-GWAS outperformed the simple evaluation based on minimal p-values (which they called MinGWAS), as well as the current state-of-the-art methods, MTAG. They then applied this C-GWAS method on a previously published GWAS meta-analysis dataset of 78 correlated facial traits. They reported previously identified loci, as well as novel loci associated with facial traits, with some validation evidence. The authors also provided an open-source R package, in which C-GWAS was implemented.

Since there are clear needs for combining GWAS summary statistics of multiple traits, this new method should be a good addition to the field. However, the method and the application parts both need some clarification or further evidence to consolidate certain claims.

1. In section 'C-GWAS performance via computer simulations' (from line 112), the authors should first provide a brief description to illustrate the motivation of the 10 simulated scenarios. I would also recommend a flowchart (or table) to clearly show the 10 scenarios in a more structured way.
2. The authors used LD-score intercepts to estimate the potential inflation in the C-GWAS face application. The same analysis should also be conducted in comparison of C-GWAS and MTAG.
3. On page 14, line 269, "Note that the threshold of 5×10^{-8} in our adjusted MinGWAS is in fact 1.6 folds more stringent than the study-wide significance threshold of 1.2×10^{-9} in the previous MinGWAS" and line 284, "Note that the threshold of 3×10^{-6} in our MinGWAS is in fact 1.15 folds more stringent than the study-wide suggestive significance threshold of 5×10^{-8} in the previous GWAS meta-analysis". Question 1: It is unclear how and by what criteria are the number 1.6 and 1.15 calculated, the authors should give a formula or description; Question 2: The strict comparison of the identified loci between two approaches should control the equal false positive rate (type I error), the authors should give some descriptions to illustrate this issue.
4. The claim of 17 novel loci needs further support, particularly when these "novel findings" are actually used to demonstrate the better performance of C-GWAS. The replication is still the gold standard for GWAS, and the current replication is quite weak (only 4 of the 17 loci were replicated). While the authors blamed the small sample size for the lack of replications, I would suggest them reach out and obtain further samples for the replication purpose. Also, the authors should check if the direction of the effect is consistent between the discovery and the replication data.
5. On page 19, line 366-373, in order to show the PRS results are consistent with phenotype results, the authors may want to add some references to describe the facial features of the main major continental groups.
6. The extended Figure 9 is not illustrative enough to understand, a simulated face of each population from the PRS results would be more reader friendly.
7. Several terms were introduced without clear definitions upon the first mentioning (e.g. `getI(G)`, `getPsi(G)`, `getCoef`).
8. The labels in several figures are too small to read (e.g. Figure 3, Figure 6a&6c, Extended Data Figure 1). These figures should be improved.

Reviewer #2 (Remarks to the Author):

In this paper, the authors develop a multi-trait GWAS method, C-GWAS, which tests whether each SNP is associated with any of the traits represented in the multiple sets of GWAS summary statistics included. Based on their simulations, they find that their approach has greater power than taking the minimum p-value of the primary GWAS or the minimum p-value of a set of summary statistics created by the MTAG method. They also find that their approach is computationally feasible in very large sets

of summary statistics. They apply C-GWAS to a large set of facial morphology GWASs from the Rotterdam Study and find several lead SNPs that would not have been discovered by the minimum P-value or MTAG p-values alone. These use these results to create and test a polygenic risk score and to conduct a series of bioinformatic follow-up analyses.

The method seems like it is potentially a powerful tool for multi-trait analysis. However, I have several concerns with the manuscript that I describe below.

1) The main text of the paper is very difficult to follow. The authors refer to formulas, scenarios, and data that are explained only in the methods section. While the methods section should have the sort of detailed information that allows readers to more deeply understand what is described in the main text, for a general interest journal like Nature Communication the main text should stand on its own and be possible to follow by non-specialists. However, in this manuscript, I don't believe that there is not enough information in the main text to understand what C-GWAS is doing and how it is being tested.

2) Relatedly, the authors sometimes use terms and variables that are not defined at all in the main text (e.g., the "inflation estimation function" on line 174 or "alpha" in line 189). Other times, the authors define variables very early in the manuscript and then use them without redefining them later in the manuscript (e.g., π and ψ). This is difficult for readers, who are not as familiar with the notation used by the authors, and small reminders describing what these variables are could be helpful. Finally, sometimes the authors use vague, non-standard terms to define variables and I wasn't sure what they meant until I looked them up in the methods section (e.g., "effect correlation" and "background correlation").

3) I was surprised that MTAG was a central comparison in the paper since the null hypothesis (and the goal) of MTAG is different than that of C-GWAS. Both are multi-trait methods, but C-GWAS tries to identify SNPs that are associated with any of the analyzed traits and MTAG tries to identify SNPs that are associated with each trait separately, and as such produces p-values for each input GWAS. The authors try to account for this by taking the minimum of the MTAG p-values, but because the authors are trying to use MTAG for something different than what it was meant for, they find that it performs poorly relative to their new approach. Why do the authors compare C-GWAS to MTAG and not to other methods with the same null hypothesis? For example, Lui et al. (2018) is a more recent method that I think would be a better comparison. They also refer to the Shom method of Zhu et al. (2015), but ignore the Shet method found in the same paper.

4) PRS for complex traits that use more SNPs tend to have greater predictive power. As such, it is not surprising that the C-GWAS-based PRS is more predictive than the minGWAS PRS since it has more SNPs. What happens if the authors use the top 57 SNPs from the minGWAS results to make the PRS more comparable to the C-GWAS-based PRS?

References

Liu, Z., & Lin, X. (2018). Multiple phenotype association tests using summary statistics in genome-wide association studies. *Biometrics*, 74(1), 165-175.

Zhu, X. et al. Meta-analysis of correlated traits via summary statistics from GWASs with an application in hypertension. *Am J Hum Genet* 96, 21-36 (2015).

We are grateful to the editor and reviewers for their insightful comments on our manuscript. We have revised the manuscript extensively and believe that our revision significantly increased the quality and readability of our manuscript, hoping that the editors and reviewers agree. In the following, we provide a summary of the major improvements based on the reviewers' comments and below the point-by-point reply to all reviewers' comments and a description how we revised the manuscript accordingly.

Summary of major improvements:

1. To strengthen the replication evidence of our newly identified genetic face loci, we have substantially increased our replication dataset from previously 1174 Europeans of one cohort to now 13,622 Europeans and Asians of 6 cohorts, which additionally allowed replication testing across two major continental populations. We additionally added data from additional 2,774 individuals of European descent from two cohorts: Pennsylvania State University (PSU) and Indiana University-Purdue University Indianapolis (IUPUI), that were published in a recent European face GWAS paper¹. In addition, we used the summary statistics from a very recent Asian face GWAS² consisting of 9,674 Chinese individuals from three cohorts: the National Survey of Physical Traits (NSPT), the Northern Han Chinese (NHC) and the Taizhou Longitudinal Study (TZL).
2. We additionally compared C-GWAS with the three statistics (*mixAda*, *mixFisher* and *mixTippett*) proposed by Liu et al.³ and the S_{Het} statistic proposed by Zhu et al.⁴ in real face data consisting of 30 traits. The comparison showed that C-GWAS significantly outperform all of these previous methods in terms of statistical power and type-I error rate, providing extra evidence on the advancement of our new C-GWAS method.
3. We carefully restructured our manuscript to make it easier for readers to follow.

Point-by-point reply to the all reviewers' comments.

Reviewer #1 (Remarks to the Author):

The authors proposed C-GWAS, a new method for combining GWAS summary statistics of multiple traits. They performed extensive simulations to show C-GWAS outperformed the simple evaluation based on minimal p-values (which they called MinGWAS), as well as the current state-of-the-art methods, MTAG. They then applied this C-GWAS method on a previously published GWAS meta-analysis dataset of 78 correlated facial traits. They reported previously identified loci, as well as novel loci associated with facial traits, with some validation evidence. The authors also provided an open-source R package, in which C-GWAS was implemented.

Since there are clear needs for combining GWAS summary statistics of multiple traits, this new method should be a good addition to the field. However, the method and the application parts both need some clarification or further evidence to consolidate certain claims.

Reply: We thank this reviewer for the positive comments and provide the requested clarifications according to which we revised our manuscript in the point-by-point reply below.

1. In section 'C-GWAS performance via computer simulations' (from line 112), the authors should first provide a brief description to illustrate the motivation of the 10 simulated scenarios. I would also recommend a flowchart (or table) to clearly show the 10 scenarios in a more structured way.

Reply: We extended our overview of C-GWAS (line 86-141) to provide concepts, terms, and formula that are necessary for readers to understand the motivation of our simulations. In addition, we added descriptions (line 143-146) in the revised Results section to summarize the motivation of our simulations.

2. The authors used LD-score intercepts to estimate the potential inflation in the C-GWAS face application. The same analysis should also be conducted in comparison of C-GWAS and MTAG.

Reply: We added this analysis in our revised manuscript (line 372-374). We used LD-score intercepts to estimate the potential deviation from the expected null in both the C-GWAS and the MTAG applications. The LD-score intercept of C-GWAS was very close to 1.0 (0.997) as expected, whereas the 30 LD-score intercepts of MTAG all showed severe deflations (mean=0.834; sd=0.042). We added these new results to the revised Results section (line 372-374).

3. On page 14, line 269, "Note that the threshold of 5×10^{-8} in our adjusted MinGWAS is in fact 1.6 folds more stringent than the study-wide significance threshold of 1.2×10^{-9} in the previous MinGWAS" and line 284, "Note that the threshold of 3×10^{-6} in our MinGWAS is in fact 1.15 folds more stringent than the study-wide suggestive significance threshold of 5×10^{-8} in the previous GWAS meta-analysis" . Question 1: It is unclear how and by what criteria are the number 1.6 and 1.15 calculated, the authors should give a formula or description; Question 2: The strict comparison of the identified loci between two approaches should control the equal false positive rate (type I error), the authors should give some descriptions to illustrate this issue.

Reply: We revised our manuscript to explain these points in more details.

Answer for question 1 (line 286-299)

For a fair comparison, we adjusted the previous MinGWAS, which is by definition inflated under the null, using our *getCoef* function. A simulation analysis imbedded in C-GWAS

confirmed that under the null, both C-GWAS and the adjusted MinGWAS tightly followed the uniform distribution (Fig. 4b-c). This means that the comparison among the two approaches is very strict under the same α . Therefore, the traditional genome-wide significance threshold of 5×10^{-8} used in a standard GWAS corresponds to our study-wide significance threshold. Note that when applying the threshold of 5×10^{-8} in the adjusted MinGWAS, which corresponded to an unadjusted p-value of 7.52×10^{-10} , i.e., the same number of SNPs survived 7.52×10^{-10} before and 5×10^{-8} after the *getCoef* adjustment, is in fact 1.6 folds more stringent than the study-wide significance threshold of 1.2×10^{-9} in the previous MinGWAS. This observation is consistent with our simulation scenario 9 showing that adjusting for MinGWAS using the estimated number of independent traits is prone to inflation under stringent thresholds.

Answer for question 2 (line 287-291)

A simulation analysis imbedded in C-GWAS confirmed that under the null, both C-GWAS and the adjusted MinGWAS tightly followed the uniform distribution (Fig. 4b-c). This means that the comparison among the two approaches is very strict under the same α .

4. The claim of 17 novel loci needs further support, particularly when these “novel findings” are actually used to demonstrate the better performance of C-GWAS. The replication is still the gold standard for GWAS, and the current replication is quite weak (only 4 of the 17 loci were replicated). While the authors blamed the small sample size for the lack of replications, I would suggest them reach out and obtain further samples for the replication purpose. Also, the authors should check if the direction of the effect is consistent between the discovery and the replication data.

Reply: Following the reviewer’s suggestions, we reached out and obtained additional 2,774 individuals of European descent from the PSU cohort (n=1,990) and the IUPUI cohort (n=784) as described previously in White et al.¹. This represents the part of the published data used by White et al that did not overlap with the data used in our initial Xiong et al. study⁵ that we used here for C-GWAS discovery. Although the full dataset of White et al had been made publicly available by the authors, because of the partial overlap between the two studies, we approached the authors responsible for these two non-overlapping studies and they provided their data to us. Because these data were previously published (together with the overlapping ones), the respective colleagues did not request co-authorship for providing their data to our current study; hence, our author list did not change. We acknowledge their help in making the non-overlapping data available to us in the revised acknowledgement section. This addition increased our European replication dataset from previously 1,174 from one cohort to now 3,948 from three cohorts. GWASs of facial variation and meta-analyses of the two added European cohorts were conducted according to the previously described protocol¹. These analyses resulted in 63 sets of GWAS summary statistics corresponding to the 63 facial segments.

In addition, we downloaded the publicly available summary statistics from a recent facial variation GWAS of Chinese population², which included 9,674 Chinese individuals

from three cohorts, i.e., the National Survey of Physical Traits (NSPT), the Northern Han Chinese (NHC), and the Taizhou Longitudinal Study (TZL). Since this dataset is publicly available as we used it, no change to the authorship was required. In the revised acknowledgement section, we acknowledge the authors of this publication for making their data publicly available. The GWASs and meta-analyses of Chinese cohorts were carried out according to the previously described protocol², which also resulted in 63 sets of GWAS summary statistics for 63 facial segments. We then carried out our replication analysis for the 56 SNPs in RS, PSU+IUPUI, NSPT+NHC+TZL, and combined result of these six cohorts.

We used Bonferroni and Fisher' s combined test to combine the evidence of replication and accounted for the trait differences between our landmarks-based study and the previous facial segment-based studies. In brief, we first adjusted the minimal p-value of the 63 segment-based p-values using Bonferroni correction of the numbers of independent traits as reported in previous studies. We then used Fisher' s combined test to combine the RS p-value derived from C-GWAS, the adjusted minimal p-value from PSU+IUPUI, and the adjusted minimal p-value from NSPT+NHC+TZL. In short, the test statistic of the Fisher' s combined test T_F of n independent tests with p-value P_n follows a chi-squared distribution of $2n$ degree of freedom as $T_F = -2 \sum \log(P_n) \sim \chi_{2n}^2$.

Replication for 56 regional lead SNPs with study-wide suggestively significant evidence was conducted in three datasets including 1,174 Dutch individuals from RS that were not previously included in the study of Xiong et al., 2,774 individuals of European descent from the Pennsylvania State University (PSU) and the Indiana University-Purdue University Indianapolis (IUPUI), and 9,674 Chinese individuals from the National Survey of Physical Traits (NSPT), the Northern Han Chinese (NHC) and the Taizhou Longitudinal Study (TZL). Combining replication evidence from all cohorts revealed an overall highly significant deviation from the expectation under the null (Kolmogorov-Smirnov test $p=9.7 \times 10^{-16}$, **Fig. 4h**). Among the 56 SNPs, 91% (51/56) showed consistent allele effects with the discovery cohort and 82% (46/56) was replicated at the nominal significance ($p < 0.05$). Focusing on the 17 novel SNPs also showed an overall significant deviation from the expectation under the null (Kolmogorov-Smirnov test $p=8.6 \times 10^{-3}$, **Fig. 4h**). 82% (14/17) showed consistent allele effects with the discovery cohort, and 53% (9/17) was replicated at the nominal significance ($p < 0.05$) (**Supplementary Table R1**).

We added these descriptions in the revised Results section “C-GWAS outperforms MinGWAS in real data of the human face” and the revised Methods section “Replication of C-GWAS findings and previous GWAS findings” and acknowledged the colleagues for having made their data available in the revised acknowledgement section.

Fig. 4h. The 56 suggestive lead SNPs from C-GWAS ($n=10,115$) were looked up in the independent combined replication dataset ($n=13,622$). SNPs passing the suggestive line are highlighted using the diamond or the cross denoting respectively identified only by C-GWAS or by both C-GWAS and MinGWAS. Previously established loci are indicated in green and novel loci in orange. Kolmogorov-Smirnov test was used to test the deviation between replicated p-values of 17 novel SNPs, all 56 SNPs and null (uniform) distribution. Solid and dashed lines denote the expected null with 95% CI.

SNP	EA	EA freq	Discovery C-GWAS P	Replication analysis				
				Effect consistency	Combined P	RS	PSU+ IUPUI	NSPT+ NHC+TZL
rs11462489	A	0.4014	2.39E-06	0.525	0.203	0.517	0.179	0.153
rs2055127	C	0.4975	8.73E-07	0.0751	0.347	0.193	0.332	0.544
rs12693444	A	0.2993	2.94E-07	0.0321	0.017	0.0311	0.0549	0.258
rs7557972	T	0.02825	5.10E-07	0.704	0.0436	0.923	0.185	0.00895
rs57839456	G	0.1515	3.32E-08	0.228	0.396	0.232	0.64	0.296
rs17278459	G	0.1243	2.50E-06	0.23	0.0466	0.981	0.0137	0.125
rs202098386	CT	0.09644	2.62E-06	0.0201	0.252	0.128	0.67	0.234
rs76828289	C	0.02143	4.13E-08	0.594	0.373	0.692	0.159	0.358
rs34485638	T	0.105	2.74E-07	0.103	0.183	0.289	0.254	0.165
rs1917407	C	0.3501	1.46E-09	0.000737	0.0588	0.14	0.113	0.146

rs72831074	G	0.03468	1.16E-06	0.346	0.591	0.502	0.782	0.25
rs174582	G	0.2156	1.34E-06	0.477	0.0329	0.114	0.409	0.0224
rs10773002	A	0.2464	2.07E-06	0.00277	0.000685	0.00984	0.0447	0.0192
rs2812241	T	0.1463	2.40E-06	0.00019	1.36E-05	0.00109	0.00376	0.0223
rs2937979	G	0.1348	7.52E-07	0.0374	0.0335	0.0896	0.0432	0.277
rs8025172	C	0.1161	2.00E-06	0.0268	0.0204	0.0196	0.498	0.057
rs8030235	T	0.3122	2.52E-06	0.0099	0.0472	0.374	0.0818	0.0557

Supplementary Table R1. 17 novel C-GWAS suggestively significant lead SNPs and their replication. EA frequency was calculated in discovery meta-analysis. Effect consistency quantiles were bold when the SNP showed consistent allele effects ($q < 0.5$). Nominal significant replicated p-values ($p < 0.05$) in combined data and in each cohort are highlighted in bold.

- On page 19, line 366-373, in order to show the PRS results are consistent with phenotype results, the authors may want to add some references to describe the facial features of the main major continental groups.

Reply: According to the comment, we added descriptions about the phenotype consistency and referred to the following references^{2,6,7} (line 415):

- Zhang, M. et al. Genetic variants underlying differences in facial morphology in East Asian and European populations. *Nat Genet* 54, 403-411 (2022)
- Noback, M.L., Harvati, K. & Spoor, F. Climate-related variation of the human nasal cavity. *Am J Phys Anthropol* 145, 599-614 (2011).
- Guo, J. et al. Variation and signatures of selection on the human face. *J Hum Evol* 75, 143-52 (2014)

- The extended Figure 9 is not illustrative enough to understand, a simulated face of each population from the PRS results would be more reader friendly.

Reply: Because our face data is limited with the 13 facial landmarks, it is difficult to have the resolution of the whole face as done in the study of Zhang et al.². To make our PRS results more reader friendly, we revised our **Extended Data Fig. 9 (Extended Data Fig. 10** in current version) to highlight the changes of face PRSs in relative to the mean PRSs of all EUR, EAS and AFR samples from the 1000 Genomes Project, as shown below.

Extended Data Figure 10. (a), A PRS analysis was carried out in the replication cohort of 1,174 individuals from the Rotterdam Study (RS). PRSs were separately constructed based on C-GWAS findings (57 lead SNPs) and MinGWAS findings (57 lead SNPs or 17 lead SNPs) for all 78 facial traits. Explained variance (R^2) and its 95% confidence intervals are presented on the left side and the significance for testing $H_0: R^2=0$ using the z-test is displayed on the right side (orange for C-GWAS, green for MinGWAS of 57 SNPs and blue for MinGWAS of 17 SNPs). (b-c), PRSs were constructed for 78 facial traits in 1,668 individuals from three major continental groups (Europe, EUR; Africa, AFR; and East Asian, EAS) of the 1000 Genomes

Project (1000G). Facial variation explainable by MinGWAS (b) and C-GWAS (c) PRS was compared with the average PRS across all continents.

- Several terms were introduced without clear definitions upon the first mentioning (e.g. getI(G), getPsi(G), getCoef).

Reply: Revised accordingly.

- The labels in several figures are too small to read (e.g. Figure 3, Figure 6a&6c, Extended Data Figure 1). These figures should be improved.

Reply: Adjusted accordingly. However, by doing so the Fig 6 became too crowded; thus, we separated the Fig into two figures (new Figure 6 and new Figure 7) as shown below.

Figure 6. Biological annotation of significant findings. (a), Gene ontology (GO), human phenotype ontology (HP), mouse phenotype ontology (MP) analyses were carried out based on C-GWAS (83 genes in 56 loci) and MinGWAS (24 gene in 17 loci) findings. Biological terms from C-GWAS and MinGWAS were plotted above and below the black solid line, respectively. Shared and significant terms are indicated in green (more significant) or blue (less significant).

Unique terms are indicated in orange with those non-significant in gray. The dashed lines indicate Bonferroni corrected $p=0.05$. A regulation network in CNCC is derived based on C-GWAS finding (**b**, 56 loci) and MinGWAS finding (**c**, 17 loci). Transcription factor (TF) - regulation element (RE) - target gene (TG) regulation triplets are superimposed on chord diagrams and are distinguished using colors. The TF-RE edge width represents the number of REs associated with TF. TGs linked to their RE are highlighted and the number of RE-TG is proportional to arrows size. Closest genes to lead SNPs are highlighted in orange text. The genes that are both TF and TG are marked using “*”.

Figure 7. Colocalization analysis of C-GWAS significant findings. Colocalization analysis was carried out for the 56 loci from C-GWAS in 22 tissues from GTEx. **(a)**, posterior probabilities in the heatmap are indicated using the box size and the gradient in color (posterior probabilities > 0.7 in solid boxes). Closest genes to the regional lead SNPs are indicated using orange text. **(b-e)**, Four multi-tissue colocalization examples are shown, with the top right representing C-GWAS results, the bottom right representing eQTL results, and the left representing the colocalization results.

Reviewer #2 (Remarks to the Author):

In this paper, the authors develop a multi-trait GWAS method, C-GWAS, which tests whether each SNP is associated with any of the traits represented in the multiple sets of GWAS summary statistics included. Based on their simulations, they find that their approach has greater power than taking the minimum p-value of the primary GWAS or the minimum p-value of a set of summary statistics created by the MTAG method. They also find that their approach is computationally feasible in very large sets of summary statistics. They apply C-GWAS to a large set of facial morphology GWASs from the Rotterdam Study and find several lead SNPs that would not have been discovered by the minimum P-value or MTAG p-values alone. These use these results to create and test a polygenic risk score and to conduct a series of bioinformatic follow-up analyses.

The method seems like it is potentially a powerful tool for multi-trait analysis. However, I have several concerns with the manuscript that I describe below.

Reply: We thank this reviewer for agreeing with us that our new method represents a powerful tool for multi-trait analysis and reply to the concerns in the following.

- 1) The main text of the paper is very difficult to follow. The authors refer to formulas, scenarios, and data that are explained only in the methods section. While the methods section should have the sort of detailed information that allows readers to more deeply understand what is described in the main text, for a general interest journal like Nature Communication the main text should stand on its own and be possible to follow by non-specialists. However, in this manuscript, I don't believe that there is not enough information in the main text to understand what C-GWAS is doing and how it is being tested.

Reply: According to the comment, we added rationale and restructured our main text to make a stand-alone story, so that the readers can follow better. This way, we increased our efforts to explain in a better way what C-GWAS is doing and how it is being tested. In particular, we extended our overview of C-GWAS (Line 86-141), which provides concepts, terms, and formula that are necessary for readers to understand the motivation of our simulations.

- 2) Relatedly, the authors sometimes use terms and variables that are not defined at all in the main text (e.g., the "inflation estimation function" on line 174 or "alpha" in line 189). Other times, the authors define variables very early in the manuscript and then use them without redefining them later in the manuscript (e.g., π and ψ). This is difficult for readers, who are not as familiar with the notation used by the authors, and small reminders describing what these variables are could be helpful. Finally, sometimes the authors use vague, non-standard terms to define variables and I wasn't sure what they meant until I looked them up in the methods section (e.g., "effect correlation" and "background correlation").

Reply: We revised our text accordingly to make it easier for the readers to follow.

- 3) I was surprised that MTAG was a central comparison in the paper since the null hypothesis (and the goal) of MTAG is different than that of C-GWAS. Both are multi-trait methods, but C-GWAS tries to identify SNPs that are associated with any of the analyzed traits and MTAG tries to identify SNPs that are associated with each trait separately, and as such produces p-values for each input GWAS. The authors try to account for this by taking the minimum of the MTAG p-values, but because the authors are trying to use MTAG for something different than what it was meant for, they find that it performs poorly relative to their new approach. Why do the authors compare C-GWAS to MTAG and not to other methods with the same null hypothesis? For example, Lui et al. (2018) is a more recent method that I think would be a better comparison. They also refer to the Shom method of Zhu et al. (2015), but ignore the Shet method found in the same paper.

Reply:

Regarding MTAG: (line 1198-1203; line 356-358; line 375-387): C-GWAS and MTAG have an important similarity in distinguishing between the effect correlation Π (true effect Ω in MTAG) and the background correlation Ψ (estimate error Σ in MTAG) with the advantage in better estimating true null, whereas other methods do not. In addition, when Π and Ψ are different, both methods provide additional statistical power, while the other methods do not. It is therefore desirable to compare the statistical performance of the two methods.

Although the H0 of C-GWAS and MTAG are different, a fair comparison was achieved by adjusting the minimum P of MTAG using the same `getCoef` function as in C-GWAS. Besides the moderate power gain of C-GWAS due to the statistical improvements as demonstrated in our simulations, the surprisingly lower power observed for MTAG in our application to real face data is additionally explained by less accurate true effect and error estimations due to (1) a mediocre sample size and (2) severe violation of MTAG's assumption. The extra multiple testing burden posed to MTAG by forcing it to test our H0 had negligible impact given that MTAG performed similarly poor even without adjusting for 30 facial traits.

Regarding other methods (line 1217-1220; line 388-398): We agree with the reviewer that the comparison with other potential competitive methods is meaningful. To this end, We additionally compared C-GWAS with four competing statistics, including *mixAda*, *mixFisher* and *mixTippett* proposed by Liu et al. using R package 'MPAT'³ and S_{Het} proposed by Zhu et al. using R script 'CPASSOC'⁴ in the same 30 facial GWASs. This analysis showed that our novel C-GWAS outperformed all these previous methods, which underlines the power of our new method. The power of S_{Het} was significantly lower than that of C-GWAS as reflected by the lower capacity in reidentifying the previously reported face loci (S_{Het} 75% vs. C-GWAS 100%, **Supplementary Table 1, Extended Data Fig. 9**). The power of the three statistics proposed by Liu et al. was very low (up to 13% re-identified

loci), likely because it ignores phenotype subsets. In addition, unlike C-GWAS, these methods all had non-negligible inflations (LD intercept > 1.04).

We added these descriptions in the revised Results section “C-GWAS outperforms MTAG in real data of the human face” and the revised Methods section “Comparison with other multi-trait analysis methods using facial data”

	C-GWAS	mixAda	mixFisher	mixTippett	S _{Het}
Total time (16 parallel threads, hour)	0.57	9.84	0.34	0.33	0.65
LD intercept	0.997	1.041	1.04	1.045	1.073
Significant SNP number (<5e-8)	649	18	18	16	252
Identified significant loci (number)	16	12	9	8	16
Reported significant loci (number (%))	16 (100%)	1 (8%)	0 (0%)	1 (13%)	12 (75%)
Suggestive SNP number (<1.35e-6)	1043	160	137	107	880
Identified suggestive loci (number)	32	61	54	35	56
Reported suggestive loci (number (%))	22 (69%)	9 (15%)	8 (15%)	4 (11%)	18 (32%)

Supplementary Table 1. Performance comparison between C-GWAS and *mixAda*, *mixFisher*, *mixTippett* and S_{Het}. For the four methods tested here for comparison which were not designed to run on parallel threads, we used R package “foreach” and “doParallel” as used in C-GWAS to parallel run them in 16 threads for acceleration.

Extended Data Figure 9. A list of 327 previously established face-associated SNPs was looked up in the C-GWAS in comparison with *mixAda*, *mixFisher*, *mixTippett* and S_{Het} . The number of SNPs passing Bonferroni correction was displayed for each method. Solid and dashed lines denote the expected null with 95% CI.

- 4) PRS for complex traits that use more SNPs tend to have greater predictive power. As such, it is not surprising that the C-GWAS-based PRS is more predictive than the minGWAS PRS since it has more SNPs. What happens if the authors use the top 57 SNPs from the minGWAS results to make the PRS more comparable to the C-GWAS-based PRS?

Reply: According to the comment, we additionally constructed PRSs using 57 lead SNPs from MinGWAS (line 407-411). These PRSs explained on average 2.03% and up to 4.33% sex- and age-adjusted facial variance (**Extended Data Fig. 10a**), which was still significantly lower than C-GWAS (on average 2.28%, up to 4.51%, paired Wilcoxon rank-sum test $p\text{-value}=9\times 10^{-4}$). We added these descriptions in the revised Results section “C-GWAS outcome increases proportion of facial phenotype variance explained” .

Extended Data Figure 10. (a), A PRS analysis was carried out in the replication cohort of 1,174 individuals from the Rotterdam Study (RS). PRSs were separately constructed based on C-GWAS findings (57 lead SNPs) and MinGWAS findings (57 lead SNPs or 17 lead SNPs) for all 78 facial traits. Explained variance (R^2) and its 95% confidence intervals are presented on the left side and the significance for testing $H_0: R^2=0$ using the z-test is displayed on the right side (orange for C-GWAS, green for MinGWAS of 57 SNPs and blue for MinGWAS of 17 SNPs). (b-c), PRSs were constructed for 78 facial traits in 1,668 individuals from three major continental groups (Europe, EUR; Africa, AFR; and East Asian, EAS) of the 1000 Genomes

Project (1000G). Facial variation explainable by MinGWAS (b) and C-GWAS (c) PRS was compared with the average PRS across all continents.

References

1. White, J.D. *et al.* Insights into the genetic architecture of the human face. *Nat Genet* **53**, 45-53 (2021).
2. Zhang, M. *et al.* Genetic variants underlying differences in facial morphology in East Asian and European populations. *Nat Genet* **54**, 403-411 (2022).
3. Liu, Z. & Lin, X. Multiple phenotype association tests using summary statistics in genome-wide association studies. *Biometrics* **74**, 165-175 (2018).
4. Zhu, X. *et al.* Meta-analysis of correlated traits via summary statistics from GWASs with an application in hypertension. *Am J Hum Genet* **96**, 21-36 (2015).
5. Xiong, Z. *et al.* Novel genetic loci affecting facial shape variation in humans. *Elife* **8**(2019).
6. Noback, M.L., Harvati, K. & Spoor, F. Climate-related variation of the human nasal cavity. *Am J Phys Anthropol* **145**, 599-614 (2011).
7. Guo, J. *et al.* Variation and signatures of selection on the human face. *J Hum Evol* **75**, 143-52 (2014).

REVIEWER COMMENTS

Reviewer #1 (Remarks to the Author):

My points have been satisfactorily addressed.

Reviewer #2 (Remarks to the Author):

I thank the authors for the large number of changes they have made to the manuscript. The new version is greatly improved and I think I understand the approach they propose much better than I did at my last reading. They continue to have very strong empirical results, and I'm optimistic that their approach will likely advance the area of multi-trait genome-wide testing. However, I still have a number of concerns about the manuscript that I outline below.

1) I still find the main text insufficiently clear at points to follow the method the authors propose. This is largely because the authors use variables that they don't define. For example, on page 5, the authors introduce a variable α without defining it. On page 6 they state that " $G \in (G_1, G_2, \dots, G_n)$ " without defining what each element of the set is. (I.e., are they vectors of associations for many different SNPs for the same trait or are they vectors of associations for many different traits for the same SNP?) On page 6, the equation on line 104 was very difficult for me to parse. It was clear enough to me that the variable t had something to do with t statistics and se had something to do with standard errors, but using the notation $(1/se)$ made me think se was a scalar, but then the inverted variable is transposed, so it must be a matrix or vector. However, if it is a matrix or vector, then how is $(1/se)$ defined? (This was further complicated since I didn't realize the Σ was a summation rather than another matrix.) At other parts of the manuscript, the authors also switch between capital and lower case variables without defining what those are. The above list isn't exhaustive. The authors should review their manuscript and methods section and clearly define any variables and functions used.

2) I appreciate all the work the authors did to test other multi-trait methods. However, the authors still give undue focus to the MTAG method in their manuscript and very little focus to methods that are more comparable. I understand their point that both C-GWAS have similar underlying models, but the fact still remains that the purpose of the methods are different. Indeed, as far as I can tell, MTAG does very poorly relative to the other methods considered that are designed to test the same null hypothesis of C-GWAS. It is difficult to do a comparison of C-GWAS, MTAG, and the new methods because the new approaches are relegated to a minor, stand-alone section. It would be good to have a discussion of the comparative performance of all of these methods together so readers to assess the value of the different approaches.

3) I was confused about whether there is a different Ψ and Π matrix for each SNP. My best understanding based on the manuscript is that you estimate a common Ψ and Π for the whole genome. This is further confirmed by the lack of a subscript on the Ψ and Π matrices. However in the paper, the authors claim that this type of homogeneity assumption is a major weakness of MTAG that they correct in C-GWAS. So that makes me worried that I have misunderstood the method.

Minor Points

4) Can the authors clarify whether what they call the "background correlation" is another name for the "sampling correlation" of the GWAS estimates? I'm more familiar with the latter term. If they are the same. It may be helpful to say this the first time you use the term.

5) On page 17, the authors claim that "C-GWAS gained 54% extra statistical power as estimated using the method described in Turley et al." I'm not sure what method they are referring to in Turley et al.

Are they looking at the increase in the mean chi2 statistics?

6) Typo: Line 580, "Under the polygenity..." Should this be "Under polygenicity"?

Point-by-point reply to all reviewers' comments.

Reviewer #1 (Remarks to the Author):

My points have been satisfactorily addressed.

Reviewer #2 (Remarks to the Author):

I thank the authors for the large number of changes they have made to the manuscript. The new version is greatly improved and I think I understand the approach they propose much better than I did at my last reading. They continue to have very strong empirical results, and I'm optimistic that their approach will likely advance the area of multi-trait genome-wide testing. However, I still have a number of concerns about the manuscript that I outline below.

1) I still find the main text insufficiently clear at points to follow the method the authors propose. This is largely because the authors use variables that they don't define. For example, on page 5, the authors introduce a variable α without defining it. On page 6 they state that $G \in (G_1, G_2, \dots, G_n)$ without defining what each element of the set is. (I.e., are they vectors of associations for many different SNPs for the same trait or are they vectors of associations for many different traits for the same SNP?) On page 6, the equation on line 104 was very difficult for me to parse. It was clear enough to me that the variable t had something to do with t statistics and se had something to do with standard errors, but using the notation $(1/se)$ made me think se was a scalar, but then the inverted variable is transposed, so it must be a matrix or vector. However, if it is a matrix or vector, then how is $(1/se)$ defined? (This was further complicated since I didn't realize the Σ was a summation rather than another matrix.) At other parts of the

manuscript, the authors also switch between capital and lower case variables without defining what those are. The above list isn't exhaustive. The authors should review their manuscript and methods section and clearly define any variables and functions used.

Reply: Revised accordingly. In summary, we checked every variable and equation and revised it if it was not defined or unclearly defined. For specific concerns raised in 1), the reply is as below: (1), the variable alpha on page 5 is firstly defined in line 64 on page 4 as type-I error rate (α). (2), Now we drop the use of “G” and directly use “K GWASs” to denote the input of C-GWAS. (3) The equation in line 104 (now in line 125) has been fully revised as below: “For each SNP, let $\boldsymbol{\sigma} = (\sigma_1, \sigma_2, \dots, \sigma_K)^T$, $\mathbf{n} = (n_1, n_2, \dots, n_K)^T$ and $\mathbf{t} = (t_1, t_2, \dots, t_K)^T$ be the vectors of standard errors, sample sizes, and T statistics from K GWASs, the combined T statistic of IVW is $t_{\text{meta}} = \mathbf{w}^T \mathbf{t} / \sqrt{\mathbf{w}^T \mathbf{w}}$, where \mathbf{w} is a vector of weights, usually we choose $\mathbf{w} = (\sqrt{n_1}, \sqrt{n_2}, \dots, \sqrt{n_K})^T$ or $\mathbf{w} = (1/\sigma_1, 1/\sigma_2, \dots, 1/\sigma_K)^T$.”. (4) Capital and lower case variables for correlation matrix ($\boldsymbol{\Psi}$ and ψ , $\boldsymbol{\Pi}$ and π) are clearly defined now in line 94-98: “Both $\boldsymbol{\Psi}$ and $\boldsymbol{\Pi}$ are K by K symmetric matrices, where Ψ_{ij} , $i \in \{1, 2, \dots, K\}$ and $j \in \{1, 2, \dots, K\}$ (in brief ψ ,) indicating the correlation between the T statistics of i -th GWAS and j -th GWAS caused only by non-genetic effects and similarly Π_{ij} (or π) indicating the correlation caused only by allelic effects.”.

2) I appreciate all the work the authors did to test other multi-trait methods. However, the authors still give undue focus to the MTAG method in their manuscript and very little focus to methods that are more comparable. I understand their point that both C-GWAS have

similar underlying models, but the fact still remains that the purpose of the methods are different. Indeed, as far as I can tell, MTAG does very poorly relative to the other methods considered that are designed to test the same null hypothesis of C-GWAS. It is difficult to do a comparison of C-GWAS, MTAG, and the new methods because the new approaches are relegated to a minor, stand-alone section. It would be good to have a discussion of the comparative performance of all of these methods together so readers to assess the value of the different approaches.

Reply: We fully incorporated the reviewer's comment and substantially revised our result section "C-GWAS outperforms MTAG and other methods in real data of the human face" as below.

Next, we compared the performance of C-GWAS and MTAG in the real face data. To this end, because MTAG could not complete the analysis of all 78 GWASs within our computational capacity, we focused on a subset of 30 GWASs largely covering the left side of the strongly symmetrical human face. Although the H_0 of C-GWAS and MTAG are different, a fair comparison was achieved by adjusting the minimum p-value of MTAG using the same *getCoef* function as in C-GWAS. In addition, we compared C-GWAS with four competing statistics, including *mixAda*, *mixFisher* and *mixTippett* proposed by Liu et al.²³ and S_{Het} proposed by Zhu et al.² (**Supplementary Table 1**). These four statistics test the same H_0 as C-GWAS, but do not distinguish Π and Ψ as C-GWAS and MTAG. Both C-GWAS and S_{Het} consider GWAS subsets

whereas others do not.

Under the study-wide significance threshold of 5×10^{-8} , C-GWAS identified 649 SNPs from 16 distinct loci, all (100%) of which represent previously established facial loci. MTAG only identified five SNPs from two loci, one (50%) of which has been previously established (**Extended Data Fig. 8a-b**). The *mixAda* appeared the best performing among the three statistics proposed by Liu et al.²³, which identified 18 SNPs from 12 loci, one (8%) of which has been previously established. The S_{Het} identified 252 SNPs from 16 loci, 12 (75%) of which overlapped with previously established loci. The 2q31.1 *MTX2* was the only locus overlapping between all methods.

Considering the study-wide suggestive threshold of 1.35×10^{-6} , C-GWAS identified 1,043 SNPs from 32 distinct loci, 22 (69%) of which represent previously established facial loci. MTAG only identified 55 SNPs from 15 loci, 3 (20%) of which has been previously established (**Extended Data Fig. 8a-b**). The *mixAda* again was the best performing statistic among those proposed by Liu et al.²³, which identified 160 SNPs from 61 loci, 9 (15%) of which have been previously established. The S_{Het} identified 880 SNPs from 56 loci, 18 (32%) of which overlapped with previously established loci. The 2q31.1 *MTX2* and 12q24.21 *TBX3* were the only loci overlapping between all methods.

The LD-score intercept of C-GWAS was very close to 1.0 as

expected, whereas the MTAG showed severe deflation (mean of 30 LD-score intercepts = 0.83; sd = 0.04). In contrary, the S_{Het} showed severe inflation (1.07) and the three statistics proposed by Liu et al. also showed non-negligible inflation (>1.04) (**Supplementary Table 1**).

Finally, we looked up the 327 previously established face-associated SNPs and it was obvious that C-GWAS p-values deviated much further from the null than did MTAG and *mixAda* p-values, the latter two were largely non-significant (**Extended Data Fig. 8c**). C-GWAS replicated considerably more SNPs (49) than did S_{Het} (33) after Bonferroni correction ($p = 2 \times 10^{-4}$), although S_{Het} p-values also significantly deviated from the null (**Extended Data Fig. 9**).

The observation that MTAG showed severe deflations and re-identified only a small number of previously established loci appears surprising and suggests that the GWAS dataset of Xiong et al. is not suitable for MTAG. Note that the extra multiple testing burden posed to MTAG by forcing it to test our H_0 had negligible impact given that MTAG performed similarly poor even without adjusting for the 30 facial traits. Complex patterns of SNP effects across different facial traits likely led to violation of the assumption in MTAG. The fact that S_{het} outperformed MTAG and *mixAda* indicates that considering GWAS subsets is important when the number of GWASs is large.

3) I was confused about whether there is a different Psi and Pi matrix

for each SNP. My best understanding based on the manuscript is that you estimate a common Ψ and Π for the whole genome. This is further confirmed by the lack of a subscript on the Ψ and Π matrices. However in the paper, the authors claim that this type of homogeneity assumption is a major weakness of MTAG that they correct in C-GWAS. So that makes me worried that I have misunderstood the method.

Reply: Both Ψ and Π are K -by- K symmetric matrices where K is equal to the number of GWAS. Ψ_{ij} (or ψ) is a number between -1 and 1 indicating the correlation between the T statistics of GWAS_i and GWAS_j caused only by non-genetic effects (background correlation). Π_{ij} (or π) is also a number between -1 and 1 indicating the correlation between the T statistics of GWAS_i and GWAS_j caused only by allelic effects (effect correlation). For one C-GWAS analysis, Ψ is derived only once and does not change. Therefore, Ψ is the same for all SNPs throughout the C-GWAS analysis. Similarly, Π is the same for all SNPs. However, different from Ψ , Π is continuously updated in each iteration of i-EbICoW so that different GWAS subsets may have different Π . For example, consider 5 GWASs with estimated Ψ and Π , the Ψ of the GWAS subset 1,2,3 used in the full process of C-GWAS is $\Psi_{[1\sim 3,1\sim 3]}$ while the Π of this subset after updated in i-EbICoW may be different from initial $\Pi_{[1\sim 3,1\sim 3]}$. The high power in presence of SNP heterogeneity is obtained via (1) allowing different Π for different GWAS subsets in i-EbICoW process and (2) using truncated Wald test (TWT) for the GWAS subsets of which GWAS pairs have small $|\pi - \psi|$ after i-EbICoW process.

To clarify this, we adjusted our method section “Estimation of ψ , $\text{getPsi}(G)$ ” and “Estimation of π , $\text{getPi}(G, \Psi)$ and h , $\text{getH}(G, \Psi, \Pi)$ ”.

Minor Points

4) Can the authors clarify whether what they call the "background correlation" is another name for the "sampling correlation" of the GWAS estimates? I'm more familiar with the latter term. If they are the same. It may be helpful to say this the first time you use the term.

Reply: We think the reviewer is referring to the "the expected sample correlation" described in LDSC3. Although conceptually similar, our background correlation is not exactly "the expected sample correlation". In fact, it corresponds to the off-diagonal elements of estimate error Σ in MTAG4, which were constructed using the "estimated intercepts" from LDSC analysis of all pairs of GWASs. To make it clearer, we revised our "Overview of C-GWAS" section as below:

"C-GWAS represents a complete solution for investigating potential multi-trait effects of SNPs based only on the summary statistics of multiple GWASs (Extended Data Fig. 1). Suppose we are given the summary statistics from K GWASs as input for joint analysis. For optimizing statistical power, it is necessary to partition the correlation matrix of the T statistics from K GWASs into two matrices that differentiate the 'effect correlation' Π caused purely by true effects from the 'background correlation' Ψ under the null. Both Ψ and Π are K by K symmetric matrices, where Ψ_{ij} , $i \in \{1, 2, \dots, K\}$ and $j \in \{1, 2, \dots, K\}$ (in brief ψ), indicating the correlation between the T statistics of i -th GWAS and j -th GWAS caused only by non-genetic effects and similarly Π_{ij} (or π) indicating the correlation caused only by allelic effects. In standard meta-analysis, the off-diagonal elements of Ψ would be zero. In C-GWAS, however, the off-diagonal of Ψ may deviate

from zero in presence of sample overlap and/or non-genetic phenotype correlations caused by shared environmental or unknown factors. C-GWAS and MTAG have an important similarity in distinguishing $\mathbf{\Pi}$ from $\mathbf{\Psi}$ (we denote true effect $\mathbf{\Omega}$ and estimate error $\mathbf{\Sigma}$ in MTAG, Both $\mathbf{\Omega}$ and $\mathbf{\Sigma}$ are also K by K symmetric matrices, ideally, $\Pi_{ij} = \Omega_{ij}/\sqrt{\Omega_{ii}\Omega_{jj}}$ and $\Psi_{ij} = \Sigma_{ij}/\sqrt{\Sigma_{ii}\Sigma_{jj}}$) as key parameters in combining GWASs. Different from MTAG that generates genome-wide p-value vectors with the same number of phenotypes, C-GWAS generates a single vector of genome-wide p-values testing for each SNP if the null is deviated (H0: absence of any effect on all traits, H1: deviation from zero for at least one trait). In addition, different from MTAG that performs joint analysis of all GWASs at once, C-GWAS use two-step design where separately employs an iterative decision-making algorithm and truncated test strategy to resolve that SNPs may have different effects in different subsets of GWASs.”

5) On page 17, the authors claim that "C-GWAS gained 54% extra statistical power as estimated using the method described in Turley et al." I'm not sure what method they are referring to in Turley et al. Are they looking at the increase in the mean chi2 statistics?

Reply: Yes. We now modified our main text as “C-GWAS gained 54% extra statistical power as estimated using the increase in the mean χ^2 statistic method described in Turley et al.4” We also clarified the details in our method section: “The gain of power for C-GWAS was estimated according to the increase in the mean χ^2 statistic method described in Turley et al.4. In brief, the power ratio between two GWASs can be

derived as $\frac{E(\chi_{GWAS1}^2) - E(\chi_{null}^2)_{GWAS1}}{E(\chi_{GWAS2}^2) - E(\chi_{null}^2)_{GWAS2}}$. In our study, $E(\chi_{C-GWAS}^2) = 1.142$ and $E(\chi_{MinGWAS}^2) = 1.1$, so that $\frac{1.142 - 0.985}{1.1 - 0.998} = 1.54$ "

6) Typo: Line 580, "Under the polygenity..." Should this be "Under polygenicity"?

Reply: Thank you for pointing this out. Revised accordingly.

References

1. Liu, Z. & Lin, X. Multiple phenotype association tests using summary statistics in genome-wide association studies. *Biometrics* 74, 165-175 (2018).
2. Zhu, X. et al. Meta-analysis of correlated traits via summary statistics from GWASs with an application in hypertension. *Am J Hum Genet* 96, 21-36 (2015).
3. Bulik-Sullivan, B. et al. An atlas of genetic correlations across human diseases and traits. *Nat Genet* 47, 1236-41 (2015).
4. Turley, P. et al. Multi-trait analysis of genome-wide association summary statistics using MTAG. *Nat Genet* 50, 229-237 (2018).

REVIEWERS' COMMENTS

Reviewer #2 (Remarks to the Author):

Thanks to the authors for their careful revision of their paper. The text is much more clear for me to follow and understand. All of my concerns have been addressed.